# The evolution of drug resistance in clinical isolates of *Candida albicans*

Christopher B Ford[1,2†], Jason M Funt[1,2†], Darren Abbey[4‡], Luca Issi[5‡],
Candace Guiducci[2], Diego A Martinez[2], Toni Delorey[2], Bi yu Li[2],
Theodore C White[6], Christina Cuomo[2], Reeta P Rao[5], Judith Berman[4,7],
Dawn A Thompson[2*], Aviv Regev[1,2,3*]

[1]Department of Biology, Broad Institute of MIT and Harvard, Cambridge, United States; [2]Broad Institute of MIT and Harvard, Cambridge, United States; [3]Department of Biology, Howard Hughes Medical Institute, Massachusetts Institute of Technology, Cambridge, United States; [4]Department of Genetics, Cell Biology and Development, University of Minnesota, Minneapolis, United States; [5]Department of Biology and Biotechnology, Worcester Polytechnic Institute, Worcester, United States; [6]School of Biological Sciences, University of Missouri at Kansas City, Kansas City, United States; [7]Department of Molecular Microbiology and Biotechnology, Tel Aviv University, Tel Aviv, Israel

**\*For correspondence:** dawnt@
broadinstitute.org (DAT);
aregev@broadinstitute.org (AR)

†These authors contributed
equally to this work

‡These authors contributed
equally as second authors

**Reviewing editor:** Emmanouil T
Dermitzakis, University of
Geneva Medical School,
Switzerland

**Abstract** *Candida albicans* is both a member of the healthy human microbiome and a major pathogen in immunocompromised individuals. Infections are typically treated with azole inhibitors of ergosterol biosynthesis often leading to drug resistance. Studies in clinical isolates have implicated multiple mechanisms in resistance, but have focused on large-scale aberrations or candidate genes, and do not comprehensively chart the genetic basis of adaptation. Here, we leveraged next-generation sequencing to analyze 43 isolates from 11 oral candidiasis patients. We detected newly selected mutations, including single-nucleotide polymorphisms (SNPs), copy-number variations and loss-of-heterozygosity (LOH) events. LOH events were commonly associated with acquired resistance, and SNPs in 240 genes may be related to host adaptation. Conversely, most aneuploidies were transient and did not correlate with drug resistance. Our analysis also shows that isolates also varied in adherence, filamentation, and virulence. Our work reveals new molecular mechanisms underlying the evolution of drug resistance and host adaptation.

## Introduction

Virtually all humans are colonized with *Candida albicans,* but in some individuals this benign commensal organism becomes a serious, life-threatening pathogen. *C. albicans* possesses an arsenal of traits that promote its pathogenicity, including phenotypic switching (*Alby and Bennett, 2009*), yeast–hyphae transition (*Kumamoto and Vinces, 2005*) and the secretion of molecules that promote adhesion to abiotic surfaces (*Chandra et al., 2001*). As a commensal, an intricate balance is maintained between the ability of *C. albicans* to invade host tissues and the host's defense mechanisms (*Kim and Sudbery, 2011*; *Kumamoto and Pierce, 2011*). Alteration of this delicate host–fungus balance can result in high levels of patient mortality (*Pittet et al., 1994*; *Charles et al., 2003*): systemic *C. albicans* infections are fatal in 42% of cases (*Wisplinghoff et al., 2003*), despite the use of antifungal therapies, and *C. albicans* is the fourth most common infection in hospitals (*Gudlaugsson et al., 2003*; *Pappas et al., 2003*). While compromised immune function contributes to pathogenesis (*Gow and Hube, 2012*), it is less clear how *C. albicans* evolves to better exploit the host environment during the course of infection.

**eLife digest** Nearly all humans are infected with the fungus *Candida albicans*. In most people, the infection does not produce any symptoms because their immune system is able to counteract the fungus' attempts to spread around the body. However, if the balance between fungal attack and body defence fails, the fungus is able to spread, which can lead to serious disease that is fatal in 42% of cases.

How does *C. albicans* outcompete the body's defences to cause disease? This is a pertinent question because the most effective antifungal medicines—including the drug fluconazole—do not kill the fungus; they only stop it from growing. This gives the fungus time to develop resistance to the drug by becoming able to quickly replace the fungal proteins the drug destroys, or to efficiently remove the drug from its cells.

In this study, Ford et al. studied the changes that occur in the DNA of *C. albicans* over time in patients who are being treated with fluconazole. Ford et al. took 43 samples of *C. albicans* from 11 patients with weakened immune systems. The experiments show that the fungus samples collected early on were more sensitive to the drug than the samples collected later.

In most cases, the genetic data suggest that the infections begin with a single fungal cell; the cells in the later samples are its offspring. Despite this, there is a lot of genetic variation between samples from the same patient, which indicates that the fungus is under pressure to become more resistant to the drug. There were 240 genes—including those that can alter the surface on the fungus cells to make it better at evading the host immune system—in which small changes occurred over time in three or more patients. Laboratory tests revealed that many of these genes are likely important for the fungus to survive in an animal host in the presence of the drug.

*C. albicans* cells usually have two genetically distinct copies of every gene. Ford et al. found that for some genes—including some that make surface components or are involved in expelling drugs from cells—the loss of genetic information from one copy, so that both copies become identical, is linked to resistance to fluconazole. However, the gain of whole or partial chromosomes—which contain large numbers of genes—is not linked to resistance, but may provide additional genetic material for generating diversity in the yeast population that may help the cells to evolve resistance in the future.

These experiments have identified many new candidate genes that are important for drug resistance and evading the host immune system, and which could be used to guide the development of new therapeutics to treat these life-threatening infections.

Two classes of antifungals in clinical use target ergosterol, a major component of the fungal cell membrane: polyenes and azoles. Polyenes (e.g., Amphotericin B) are used sparingly due to toxicity (*Rex et al., 1994*), whereas azoles (e.g., fluconazole) are used widely because they can be administered orally and have few side effects (*Rex et al., 2003*). However, resistance to the azoles arises within the commensal population of the treated individual, primarily because azoles are fungistatic (inhibit growth but do not kill) (*Cowen et al., 2002*). Epidemiological data suggest that the intensity of fluconazole use is driving the appearance of resistant isolates (*Pfaller et al., 1998*). Studies of clinical isolates of *C. albicans* suggest that drug resistance can increase during an infection through the acquisition of aneuploidies (*Selmecki et al., 2009*) due to genomic plasticity and rapid evolutionary selection during infection.

Previous studies have identified two molecular mechanisms of azole resistance in *C. albicans*. First, increased activity or level of the enzymes of the ergosterol pathway (e.g., *ERG11*) reduces direct impact of the drug on its target (*Asai et al., 1999*; *Oliver et al., 2007*). Second, increased efflux of the drug from cells by ABC transporters (encoded by *CDR1* and *CDR2*) (*Coste et al., 2006*) or by the major facilitator superfamily efflux pump (encoded by *MDR1*) (*Dunkel et al., 2008*) reduces the effective intracellular drug concentration. In both cases, such alterations can result from point mutations in genes encoding these proteins (*Marichal et al., 1999*), in transcription factors that regulate mRNA expression levels (*MacPherson et al., 2005*; *Coste et al., 2006*; *Dunkel et al., 2008*), or from increased copy number of the relevant genes, via genome rearrangements such as whole chromosome and segmental aneuploidies (*Selmecki et al., 2006*; *2008*; *2009*). Indeed, the genomes of drug-resistant

strains isolated following clinical treatment often exhibit large-scale changes, such as loss of heterozygosity (LOH) (*Coste et al., 2006*; *Dunkel and Morschhauser, 2011*), copy-number variation (CNV), including short segmental CNV, and whole chromosome aneuploidy (*Selmecki et al., 2010*) accompanied by point mutations.

While we understand some aspects of the molecular basis of resistance, we understand less about the mechanisms that drive the evolution of drug resistance and overall pathogenicity in *C. albicans*. It is challenging to use forward genetic approaches in *C. albicans* due to its diploid genome and lack of a complete sexual cycle. Although *C. albicans* has conserved the genomic elements needed for mating, mating occurs instead through rare mating-competent haploids (*Hickman et al., 2013*) or via a parasexual cycle consisting of mating of diploid strains to form tetraploids followed by chromosome loss to regenerate diploids (*Bennett and Johnson, 2005*). An alternative approach is to use isolates sampled consecutively from the same patient to study the changes in the frequency of variants in natural populations under selection for drug resistance. Studies in evolved isolates have implicated multiple mechanisms in drug resistance, but have focused on large-scale aberrations such as aneuploidies and LOH (*Selmecki et al., 2008*; *2009*) or candidate genes (*Perea et al., 2001*; *White et al., 2002*), and do not comprehensively chart the genetic basis of adaptation.

Here, we used genome sequencing of isolates sampled consecutively from patients that were clinically treated with fluconazole to systematically analyze the genetic dynamics that accompany the appearance of drug resistance during oral candidiasis in human HIV patients. Most isolates from each individual patient were highly related, suggesting a clonal population structure and facilitating the identification of variation. Because each clinical sample was purified from a single colony, we cannot assess the population structure at any single time point. Instead, we have measured the occurrence of single-nucleotide polymorphisms (SNPs), CNV, and LOH events in each isolate and then compared them between isolates from the same patient and across patients' series. Consistent with previous studies, we found that LOH events were recurrent across patients' series and were associated with increased drug resistance. To identify SNPs with likely functional impact in the context of substantial genetic diversity, we focused on those events that were both persistent across isolates within a patient and were recurrent in the same gene across multiple patient series. We found 240 genes that recurrently contain persistent SNPs, many of which may be related not only to antifungal exposure but also to the complex process of adaptation to the host and antifungal exposure. In contrast, aneuploidies were prevalent in the isolates, yet they were more likely to be transient, and aneuploidy, per se, did not correlate with changes in drug resistance. Our work uses comparative analysis of a fungal pathogen to reveal new molecular mechanisms underlying drug resistance and host adaptation and provides a general model for such studies in other eukaryotic pathogens.

## Results

### Whole genome sequencing of 43 serial clinical isolates from 11 patients

To study the in vivo evolution of azole resistance in *C. albicans*, we analyzed 43 longitudinal isolates from 11 HIV-infected patients with oropharyngeal candidiasis (*White, 1997a*; *Perea et al., 2001*) (*Table 1*). The isolates were previously collected during incidences of infection and form a time series from each patient (2–16 isolates per series; *Figure 1*, *Figure 2A*). Each isolate was derived from a single colony, and thus, represents a single diploid genotype sampled from the within-host *C. albicans* population at the respective time point. In each series, the first isolate ('progenitor') was collected prior to any treatment with azole antifungals and the remaining isolates were collected at later, typically consecutive, time points, culminating in the final 'endpoint' isolate (*Table 1*).

The progenitor isolates were more sensitive to fluconazole than subsequent isolates, as defined by the minimum inhibitory concentration (MIC) (*Table 1*, 'Materials and methods'). Previous studies with some of these patient isolates identified several genomic alterations that may contribute to azole resistance, including segmental aneuploidy (*Selmecki et al., 2006*), and LOH across large chromosomal segments (*Coste et al., 2006*; *Dunkel et al., 2008*), as well as targeted alterations including increased expression of drug efflux genes (*Coste et al., 2006*), mutations in ergosterol biosynthetic genes (*Asai et al., 1999*; *Oliver et al., 2007*), and buffering by the chaperone heat shock protein 90 (Hsp90) (*Cowen and Lindquist, 2005*).

**Table 1.** Isolate history and sequencing summary

| Publication name | PT | Strain | Entry date | Drug treatment | Dose (mg/day) | E-test MIC (ug/mL) | Depth of coverage | Reads | Percent aligned |
|---|---|---|---|---|---|---|---|---|---|
| White, T.C. | 1 | 1 | 9/10/90 | Fluconazole | 100 | 0.25 | 111.96 | 9,896,468 | 87.17% |
| | | 2 | 12/14/90 | Fluconazole | 100 | 1 | 69.20 | 12,797,328 | 87.43% |
| | | 3 | 12/21/90 | Fluconazole | 100 | 4 | 92.04 | 16,987,814 | 86.87% |
| | | 4 | 12/31/90 | Fluconazole | 100 | 3 | 80.69 | 14,858,710 | 87.81% |
| | | 5 | 2/8/91 | Fluconazole | 100 | 4 | 110.80 | 20,484,584 | 86.75% |
| | | 6 | 2/22/91 | Fluconazole | 100 | 4 | 101.94 | 18,837,954 | 86.63% |
| | | 7 | 3/25/91 | Fluconazole | 100 | 4 | 81.65 | 15,123,020 | 86.66% |
| | | 8 | 4/8/91 | Fluconazole | 100 | 4 | 112.53 | 20,778,562 | 86.64% |
| | | 9 | 6/4/91 | Fluconazole | 100 | 4 | 113.18 | 22,223,228 | 83.20% |
| | | 11 | 7/15/91 | Fluconazole | 100 | 4 | 53.28 | 9,896,468 | 87.17% |
| | | 12 | 11/26/91 | Fluconazole | 200 | 4 | 96.10 | 18,282,472 | 85.54% |
| | | 13 | 12/13/91 | Fluconazole | 400 | 32 | 123.67 | 22,070,518 | 89.13% |
| | | 14 | 1/28/92 | Fluconazole | 400 | 24 | 98.66 | 18,114,916 | 87.41% |
| | | 15 | 2/21/92 | Clotriminazole | 50 | 24 | 120.90 | 22,401,374 | 86.57% |
| | | 16 | 4/1/92 | Fluconazole | 400 | 96 | 87.44 | 16,061,560 | 87.17% |
| | | 17 | 8/25/92 | Fluconazole | 800 | 96 | 97.83 | 18,317,118 | 85.91% |
| Perea, S. et al. | 7 | 412 | 2/15/95 | Fluconazole | 0 | 0.25 | 93.15 | 17,417,588 | 86.69% |
| | | 2307 | 11/22/95 | Fluconazole | 400 | 0.75 | 95.79 | 18,014,242 | 85.25% |
| Perea, S. et al. | 9 | 1002 | 4/20/95 | Fluconazole | 100 | 0.125 | 188.49 | 34,834,970 | 86.74% |
| | | 2823 | 4/6/96 | Fluconazole | 800 | | 282.62 | 52,839,288 | 86.30% |
| | | 3795 | 2/26/97 | Fluconazole | 800 | 128 | 77.63 | 13,901,062 | 88.78% |
| Perea, S. et al. | 14 | 580 | 3/13/95 | Fluconazole | 0 | 1.5 | 77.08 | 14,711,804 | 85.00% |
| | | 2440 | 1/3/96 | Fluconazole | 800 | 1.5 | 82.93 | 15,446,882 | 85.69% |
| | | 2501* | 1/4/96 | Fluconazole | 800 | 96 | 88.59 | 17,480,274 | 81.98% |
| Perea, S. et al. | 15 | 945 | 4/14/95 | Fluconazole | 300 | 4 | 108.59 | 20,591,044 | 85.19% |
| | | 1619 | 7/11/95 | Fluconazole | 500 | 64 | 93.14 | 17,565,080 | 84.69% |
| Perea, S. et al. | 16 | 3107 | 6/5/96 | Fluconazole | 800 | 4 | 97.01 | 18,361,266 | 84.84% |
| | | 3119 | 6/5/96 | Fluconazole | 800 | 96 | 87.92 | 16,615,462 | 84.67% |
| | | 3120 | 6/5/96 | Fluconazole | 800 | 96 | 105.95 | 19,442,016 | 86.79% |
| | | 3184 | 7/1/96 | Fluconazole | 800 | | 101.89 | 18,487,462 | 87.50% |
| | | 3281 | 7/16/96 | Fluconazole | 800 | | 76.44 | 14,327,376 | 85.69% |
| Perea, S. et al. | 30 | 5106 | 1/7/98 | Fluconazole | 800 | 0.5 | 87.21 | 16,466,524 | 84.67% |
| | | 5108 | 1/7/98 | Fluconazole | 800 | 0.75 | 82.32 | 17,480,274 | 81.98% |
| Perea, S. et al. | 42 | 1691 | 8/3/95 | Fluconazole | 100 | | 122.60 | 22,072,562 | 88.38% |
| | | 3731 | 12/27/96 | Fluconazole | 400 | 256 | 119.90 | 21,436,034 | 88.72% |
| | | 3733 | 12/27/96 | Fluconazole | 400 | 256 | 95.51 | 17,295,888 | 88.00% |
| Perea, S. et al. | 43 | 1649 | 7/19/95 | Fluconazole | 0 | 0.125 | 102.10 | 19,545,530 | 84.08% |
| | | 3034 | 5/15/96 | Fluconazole | 400 | 0.75 | 92.97 | 17,300,040 | 85.64% |
| Perea, S. et al. | 59 | 3917 | 2/19/97 | Fluconazole | 800 | 2 | 113.27 | 21,549,704 | 83.86% |
| | | 4617 | 8/28/97 | Fluconazole | 400 | 64 | 75.37 | 15,242,904 | 81.42% |
| | | 4639 | 9/2/97 | Fluconazole | 400 | 128 | 115.32 | 25,468,190 | 75.69% |
| Perea, S. et al. | 64 | 4018 | 4/2/97 | Fluconazole | 200 | | 110.16 | 20,118,736 | 86.78% |
| | | 4380 | 7/14/97 | Fluconazole | 200 | | 18.03 | 20,970,946 | 9.26% |

*Table 1. Continued on next page*

*Table 1. Continued*

Strains and coverage.
(Red) Not clonally derived from progenitor.
*isolated on same day from same patient as previously published strain, 2500.

We sequenced the genomic DNA of the isolates as well as the *C. albicans* lab strain, SC5314, using Illumina sequencing (53-283X coverage, 103X on average, 'Materials and methods', *Table 1*) and identified in each series point mutations, LOH events and aneuploidies that were not present in the first strain in that series. By convention, all mutations were defined relative to SC5314, the *C. albicans* genome reference strain. We validated our pipeline for detection of point mutations using Sequenom iPlex genotyping (*Storm et al., 2003*) ('Materials and methods'). We interrogated 1973 SNPs in 27 isolates from nine clinical series and found that the iPlex base calls matched 1853 (93.9%, *Figure 1—figure supplement 1A*, *Table 2*) of the calls from our computational analysis of the sequencing data. Evaluation of the discordant sites showed somewhat lower quality scores by certain metrics but did not identify any metrics that could be used to systematically revise filtering in our computational pipeline without a radical reduction in sensitivity (*Figure 1—figure supplement 1B–G*).

## Most series are clonal, but there is significant genetic diversity between isolates

We designated as *background* those polymorphisms those that are common to all isolates in a series, including the first ('progenitor') isolate (*Figure 1A*, purple) and use them to determine that isolates within most series were clonally related, suggesting a single (primary) infection source (*Figure 1B,C*, *Figure 2*, *Figure 2—figure supplement 1*, 'Materials and methods'). To distinguish between a single primary (clonal) infection (*Figure 2A*, top) and repeated, independent infections (*Figure 2A*, bottom), we determined the distance between every two isolates based on their SNP profile and used as a heuristic a neighbor-joining algorithm to construct a phylogenetic tree from this distance metric ('Materials and methods', *Figure 2B*). Patient 64 contained one *C. albicans* isolate (4018) and one *C. dubliniensis* isolate (4380); therefore, we have excluded this series from further analysis. Additionally, we detected at least one non-clonal *C. albicans* isolate in three of the remaining ten patient series (PT 9,16, 42; *Figure 2B*, red), indicating that at least ~36% of the 11 patients sampled carried more than one unrelated *Candida* strain. We removed the four non-clonal samples (*Figure 2B*, red) from further consideration, and all subsequent analyses focused on samples from the 10 patients with at least two clonal isolates.

Despite these clonal relationships, the distance between isolates indicated significant genetic diversity *within* each patient series (*Figure 2B*), typically with each isolate differing by several thousand SNPs from its 'progenitor' isolate (*Figure 2—source data 1*). These data are consistent with two different evolutionary scenarios: accumulation of de novo mutations followed by selection (*Figure 1B*), or selection acting on pre-existing variation to vary the frequency of different genotypes in the population (*Figure 1C*). The large number of SNPs detected suggests that isolates from later time points in a series are not simply direct descendants of the earlier isolate; however, since mutation and mitotic recombination rates can be elevated under stressful conditions (e.g., drug treatment *Galhardo et al., 2007*; *Forche et al., 2011*), we cannot rule out the possibility that some of the variation may be due to de novo events occurring between time points. Formally distinguishing between these two models is not possible with the samples and data at hand. However, the role of pre-existing diversity is supported by the observation that different isolates collected on the same day from the same patient (patient 14 [2440 and 2501] and patient 16 [3107 and 3119]) differed by 9668 and 18,291 SNPs, respectively (*Figure 2—source data 1*) and had very different fluconazole MIC levels (*Table 1*) and different fitness phenotypes (see below), although in each case the strains were clearly genetically related (*Figure 2B*). Thus, we conclude that a population of related but divergent genotypes of the same lineage exists within a given patient. We next sought to identify potentially adaptive genetic changes by focusing on large-scale events (LOH and aneuploidies) as well as single-nucleotide polymorphisms.

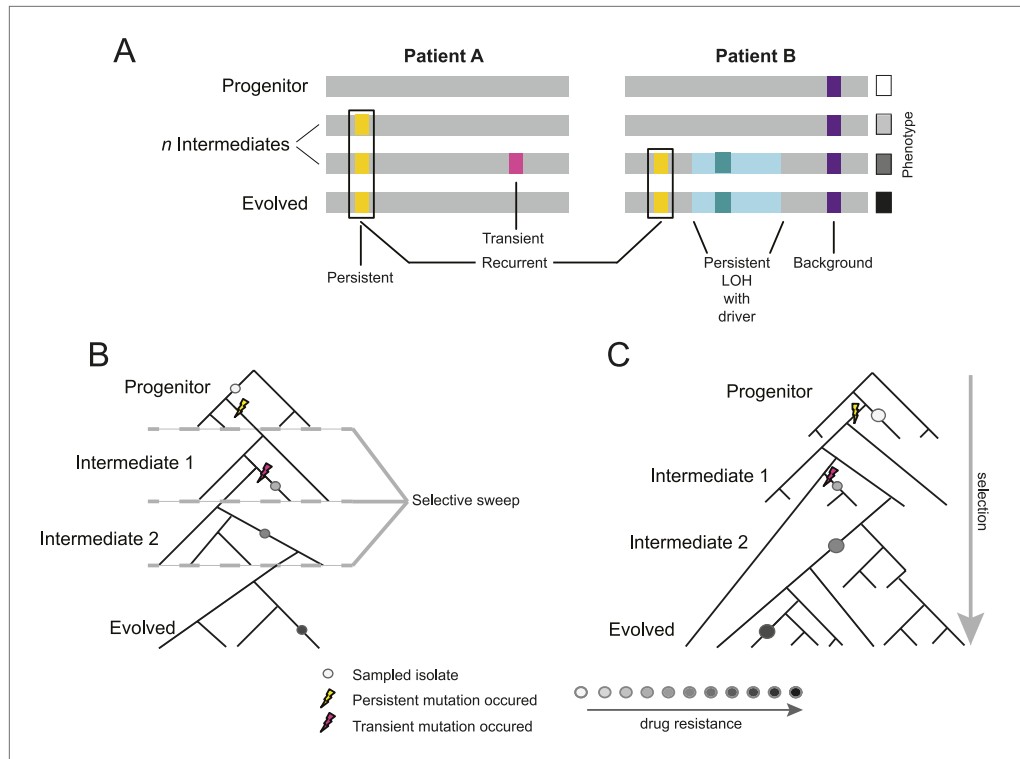

**Figure 1**. Overview of study design. (**A**) Background, persistent, transient, recurrent, and driver mutations in patient time courses. Shown is a schematic illustration of the genomes of isolates (gray bars) from two patient time courses (Patient A and B, left and right panels, respectively), ordered from the first isolate (progenitor, top) to the last (evolved, bottom). Background mutations (purple) exist in the all isolates; persistent mutations (yellow) are not in the progenitor, but found in all subsequent isolates after their first occurrence; transient mutations (pink) are not in the progenitor and only in some later isolates; recurrently polymorphic genes contain persistent mutations that occur in the same gene in more than one patient (black box). LOH events were also evaluated for persistence (light teal bar). Driver mutations, where a new persistent homozygous allele appears (e.g., G/T > A/A), are annotated in association with persistent LOH events (dark teal) and independent of these events (not shown). Each of these can be associated with a change in phenotype, such as drug resistance (boxes, right). (**B**) Sampling in the context of de novo mutation and selection bottlenecks. Each strain is a single clone (circle) isolated from an evolving population (represented by a phylogenetic tree). The population evolves and undergoes selective sweeps (dashed lines), with phenotypic changes occurring during the course of infection and treatment (i.e., drug resistance, black: high, white: low; gray scale at bottom). Persistent mutations (yellow lightning bolt) have likely swept through the population, whereas transient mutations (pink lightning bolt) have not. (**C**) Sampling in the context of selection on existing variation. Selection acts to vary the frequency of different pre-existing genotypes in the population. Persistent mutations (yellow lightning bolt) have risen in the population to a frequency that they are repeatedly sampled (large circles) whereas transient mutations (pink lightning bolt) have not (small circle).

The following figure supplement is available for figure 1:

**Figure supplement 1**. Analysis of discordant sites.

## Genetic alterations absent from the progenitor isolate, persistent within a patient, and recurrent across patients are likely adaptive

Given the high number of SNPs, LOH events and aneuploidies, we next devised a strategy to identify those changes that are more likely to play an adaptive role in drug resistance and host adaptation. We previously filtered all *background* polymorphisms, defined as any SNP relative to the reference present in all isolates from a series. Next, we defined alterations as *persistent* if present within the same patient at all subsequent time points after the 'non-progenitor' isolate in which they are first identified. We reasoned that such persistent changes will include those variants that were driven to sufficiently high frequency by selection to ensure repeated sampling (*Figure 1B,C*, yellow lightning

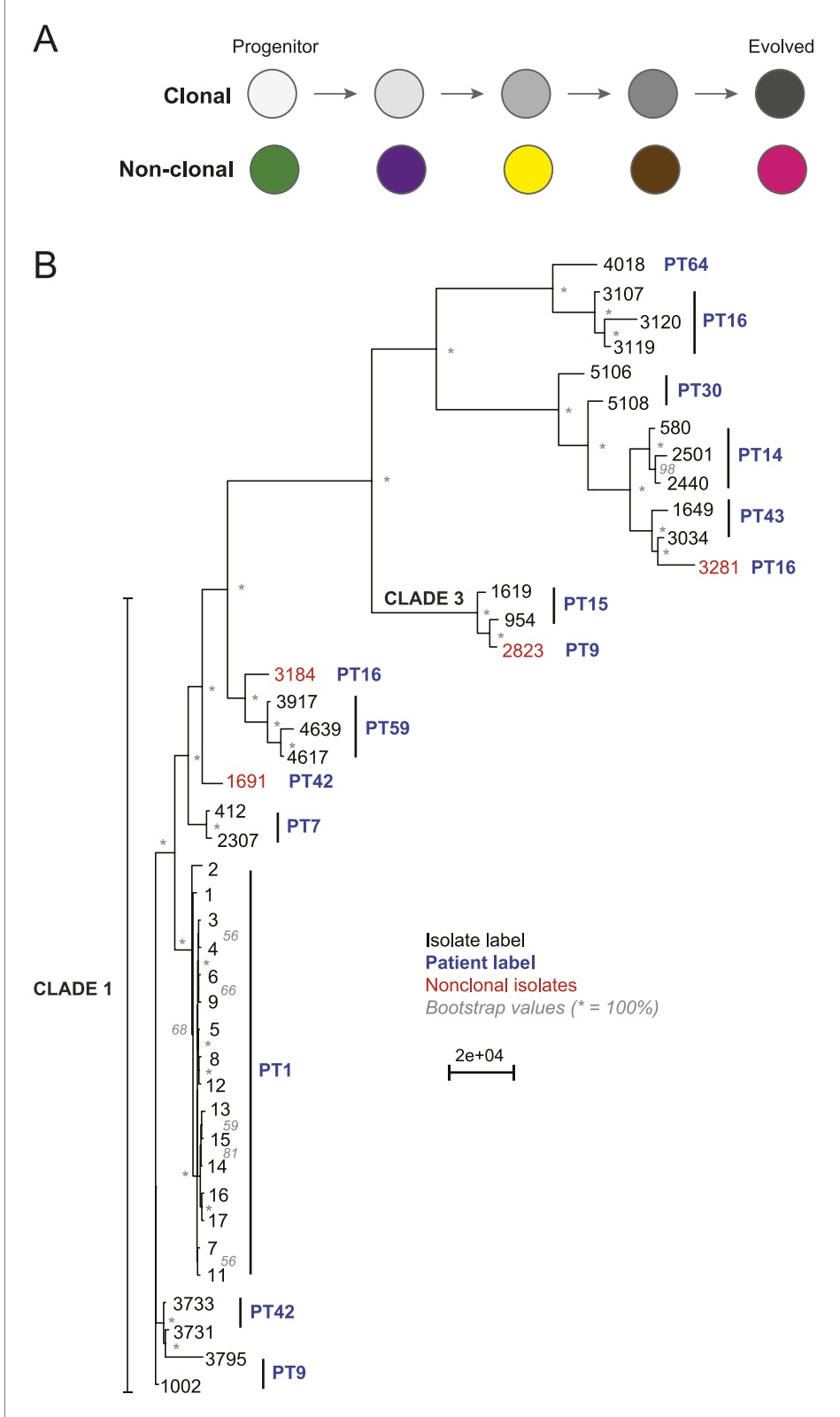

**Figure 2**. Most isolates from the same patient are clonal. (**A**) Two possible models of infection may underlie serial isolates. In the 'clonal model' (top) each subsequent sample (circle) is related to the other isolates. In the non-clonal model (bottom) isolates in a series are un-related. (**B**) The phylogenetic relationship of the isolates (black) from 11 patients (blue) was inferred based on 201,793 informative SNP positions using maximum parsimony in PAUP*.

*Figure 2. Continued on next page*

*Figure 2. Continued*

Isolates from the same patient separated by a branch distance greater than 20,000 were considered non-clonal (3281, 2823, 3184, 1691, red). Most nodes were supported by 100% of 1000 bootstrap replicates (indicated by *), expect as indicated (in gray). Clade identifiers were included as appropriate.
The following source data and figure supplement are available for figure 2:

**Source data 1**. (A) SNP category summary and all patient-series SNPs SNP category summary.
**Figure supplement 1**. SNP heterozygosity profiles for each strain.

bolt), whereas non-persistent (transient) ones do not (*Figure 1B,C*, pink lightning bolt). We consider the special case of a genetic change detected only in the endpoint isolate as 'persistent' as well, since several of the time courses consist of only two or three isolates. We apply the persistence filter to better identify potentially adaptive aneuploidies, LOH events, and SNPs.

Next, we further focused on non-synonymous polymorphisms in coding regions and employed two different strategies to identify potentially adaptive changes. In the first strategy, to identify potential *drivers* of adaptation, we focused on non-synonymous SNPs that were homozygous for a genotype not found in the progenitor strain that persisted in the subsequent isolates (e.g., G/T > A/A) consistent with positive selection. In the second strategy, we analyzed genes that were *recurrently* polymorphic across patients, such that persistent, non-synonymous polymorphisms appeared within the same open reading frame (ORF) in different patient series (*Figure 1A* and *Figure 5—source data 1A*). For recurrence, we considered only those that were not included in LOH regions, as these regions artificially inflate the estimates of persistence and recurrence. Recurrence allows us to better handle polymorphisms from the endpoint isolate in a series for which 'persistence' does not provide a meaningful filter. Thus, we further considered polymorphisms occurring only in the terminal isolate in one patient if polymorphisms also recurred in the same ORF in a series from two other patients. For example, filtering for both persistence and recurrence across at least three series reduced the number of polymorphisms for patient 1 from 13,562 polymorphisms in 5022 genes to 23 recurrent genes (*Figure 2—source data 1*, *Figure 5—source data 1A*).

## LOH events are commonly associated with increased resistance

LOH events were detected in all of the series and were often persistent, recurrent, and associated with increased drug resistance (*Figures 3 and 4*, *Figure 3—source data 1*). For example, three of four LOH events in Patient 1 were persistent and associated with an increase in MIC and both of these events were recurrent, such that LOH events in these genomic regions coincided with increases in MIC in other patients. Highly recurrent LOH events occurred on the right arm of chromosome 3 (in Patients 1, 9, 14, 16, 42, and 59; *Figure 3A*, *Figure 4A,B,D,F,H*, *Figure 3—source data 1*) and on the left arm of chromosome 5 (in Patients 1, 14, 15, and 43; *Figure 3A*, *Figure 4B,C,G*, *Figure 3—source data 1*). These regions include key genes implicated in drug resistance: on Chromosome 3, genes encoding the Cdr1 and Cdr2 efflux pumps and the Mrr1 transcription factor that regulates the Mdr1 major facilitator superfamily efflux pump (*Schubert et al., 2011*), and on Chromosome 5, genes encoding the drug target Erg11, and Tac1, a transcription factor that positively regulates expression of *CDR1* and *CDR2* (*Coste et al., 2006*). The extent of persistence and recurrence of these two LOH events is statistically significant under a naïve binary model (p < 5 × 10⁻⁴ for the Chr3R LOH; p < 0.01 for the Chr5L LOH). The recurrence of LOH events that coincide with changes in MIC suggests that they have been positively selected to rise in frequency relative to the progenitor strain. Notably, some of the recurrent LOH events may have been difficult to detect previously on SNP arrays (*Forche et al., 2008*; *Forche et al., 2004*; *Forche et al., 2005*) due to the relative paucity of SNPs in those regions in the reference strain, SC5413, itself a clinical isolate.

Putative driver mutations (defined above as non-synonymous SNPs that were homozygous for a genotype not found in the progenitor strain that persisted in the subsequent isolates; e.g., G/T > A/A) in these regions are suggestive of a point mutation followed by an LOH of the mutant allele that confers an advantage. There were 131 such mutations in 86 ORFs from 18 LOH regions from the 10 clonal patient series (*Table 1* and *Figure 5*; *Figure 5—source data 1B,C*). Some of the SNPs were in genes

**Table 2.** Sequenom iPLEX genotyping assay validation

| Patient | Isolate | Total discordant | Total concordant | Total Assayed | % Concordant |
|---------|---------|------------------|------------------|---------------|--------------|
| Patient_1 | TWTC1 | 2 | 31 | 33 | 93.94% |
| Patient_1 | TWTC2 | 1 | 32 | 33 | 96.97% |
| Patient_1 | TWTC3 | 1 | 32 | 33 | 96.97% |
| Patient_1 | TWTC12 | 1 | 32 | 33 | 96.97% |
| Patient_1 | TWTC13 | 1 | 32 | 33 | 96.97% |
| Patient_1 | TWTC15 | 1 | 32 | 33 | 96.97% |
| Patient_1 | TWTC16 | 1 | 31 | 32 | 96.88% |
| Patient_1 | TWTC17 | 1 | 32 | 33 | 96.97% |
| Patient_7 | 412 | 3 | 60 | 63 | 95.24% |
| Patient_7 | 2307 | 4 | 59 | 63 | 93.65% |
| Patient_9 | 1002 | 16 | 96 | 112 | 85.71% |
| Patient_9 | 3795 | 9 | 103 | 112 | 91.96% |
| Patient_14 | 580 | 3 | 49 | 52 | 94.23% |
| Patient_14 | 2440 | 2 | 27 | 29 | 93.10% |
| Patient_14 | 2501 | 3 | 33 | 36 | 91.67% |
| Patient_15 | 945 | 8 | 121 | 129 | 93.80% |
| Patient_15 | 1619 | 10 | 120 | 130 | 92.31% |
| Patient_16 | 3107 | 2 | 51 | 53 | 96.23% |
| Patient_16 | 3119 | 3 | 50 | 53 | 94.34% |
| Patient_16 | 3120 | 2 | 50 | 52 | 96.15% |
| Patient_30 | 5106 | 3 | 215 | 218 | 98.62% |
| Patient_30 | 5108 | 19 | 204 | 223 | 91.48% |
| Patient_43 | 1649 | 7 | 89 | 96 | 92.71% |
| Patient_43 | 3034 | 8 | 88 | 96 | 91.67% |
| Patient_59 | 3917 | 3 | 62 | 65 | 95.38% |
| Patient_59 | 4617 | 2 | 63 | 65 | 96.92% |
| Patient_59 | 4639 | 4 | 59 | 63 | 93.65% |
| TOTAL | | 120 | 1853 | 1973 | 93.92% |

that encode proteins with key known roles in drug resistance and were associated with large LOH events. For example, a nonsynonymous homozygous change in the fluconazole drug target *ERG11* was associated with the formation of the persistent LOH on the left arm of chromosome 5 in Patient 1 (*Figure 3A*), consistent with previous reports (*White, 1997b*), as was *TAC1* in Patient 42. In another example, the persistent and recurrent LOH on the right arm of chromosome 3 in Patients 9 and 16 (*Figure 4A,D*) was associated with the presence of a homozygous mutation in *MRR1* (*Schubert et al., 2011*), a regulator of *MDR1* expression. Other mutations were in genes not previously related to fluconazole resistance, including cell adhesion (*ALS3,5* and *7* and *HYR3*; [*Hoyer et al., 1998*; *Sheppard et al., 2004*; *Hoyer et al., 2008*]), filamentous growth (*FGR14, FGR28,* and *EFH,* [*Uhl et al., 2003*; *Connolly et al., 2013*]), and biofilm formation (*BCR1* and *YAK1*; [*Nobile and Mitchell, 2005*; *Goyard et al., 2008*; *Noble et al., 2010*; *Nobile et al., 2012*]). Thus, the detection of known genes involved in drug resistance confirms the approach works and that detection of genes involved in processes implicated in virulence, suggests that these process are co-evolving.

## Aneuploidies are not predictive of MIC, but may facilitate the appearance of drug resistance

Aneuploidies, either whole chromosomal or segmental, were evident in at least one isolate from 80% (8/10) of the clonal patient series, with the most prevalent aneuploidies involving Chromosome 5

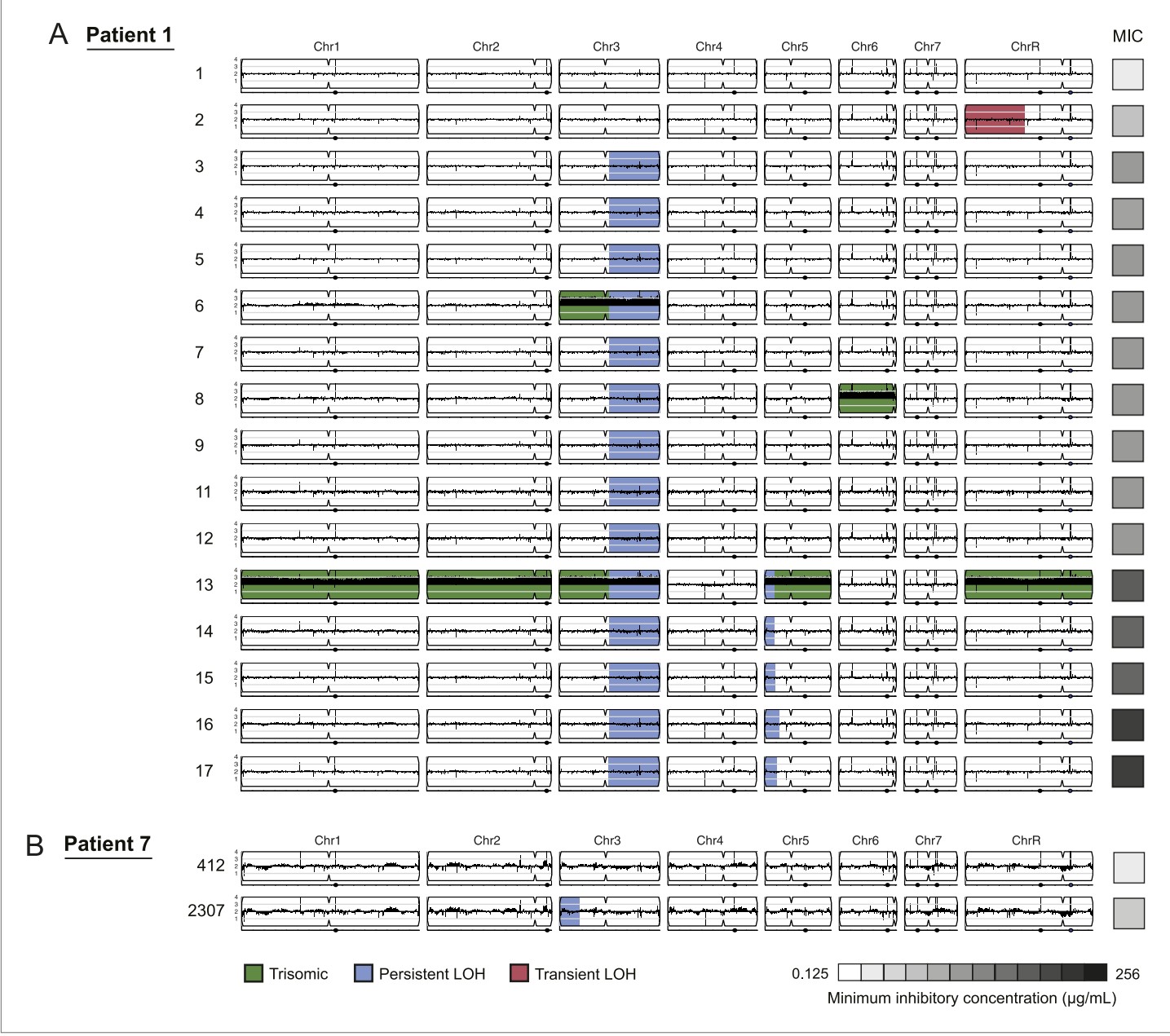

**Figure 3**. LOH events were often persistent while aneuploidies were often transient. For each time series shown are the genomes of all isolates (rows) from a patient, ordered from the first isolate (progenitor, top) to the last (evolved, bottom). Boxes on right indicate the MIC of the respective strain (black: high, white: low; gray scale at bottom). Persistent LOHs: blue, transient LOHs: pink; trisomies (all transient): green. The sequence coverage along each chromosome is indicated by black tickmarks. (**A**) Patient 1 has four LOH events, each coinciding with an increase in MIC (gray scale boxes, right). One LOH is transient (isolate 2, chromosome R, pink) and three are persistent (isolate 3, chromosome 3; isolate 13, chromosome 5; and isolate 16, chromosome 5, blue). The ploidy changes (isolates 6, 8, 13) are all transient. (**B**) Patient 7 has one LOH event (isolate 2307, chromosome 3, blue) which coincides with an increase in MIC.

The following source data is available for figure 3:

**Source data 1**. Persistent LOH regions LOH map.

(6 of 8 patients with at least one aneuploid isolate; *Figure 3A* and *Figure 4B–G*, green). In contrast to the persistence of most LOH events, persistent aneuploidies were rarer and were not consistently associated with adaptive increases in MIC levels (*Figures 3 and 4*). This is consistent with the

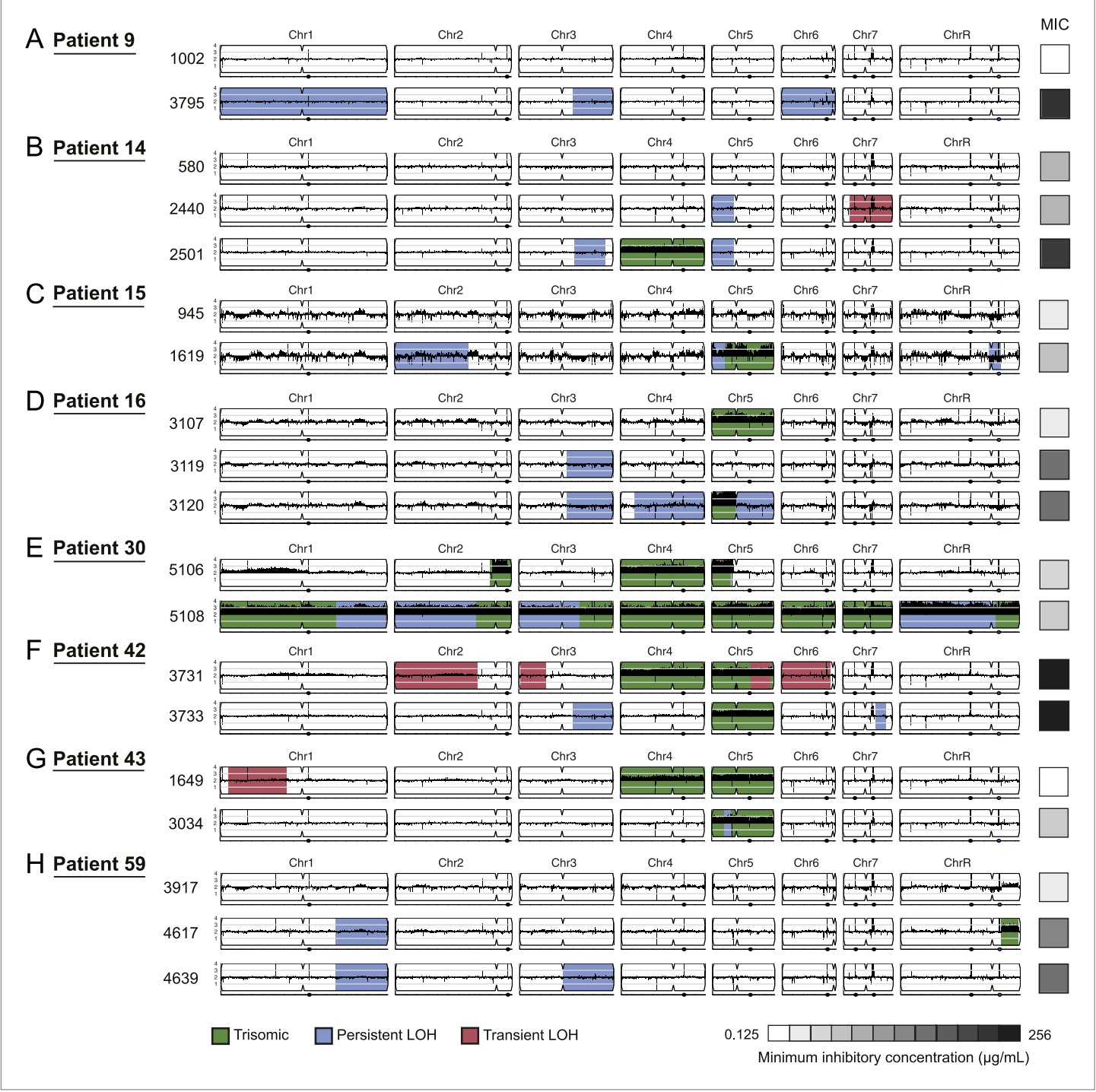

Figure 4. Persistent and transient LOH and aneuploidies. For each time series shown are the genomes of all isolates (rows), ordered from the first isolate (progenitor, top) to the last (evolved, bottom). Boxes on right indicate the MIC of the respective strain (black: high, white: low, gray scale at bottom). Persistent LOHs: light blue, transient LOHs: pink; trisomies (all transient): green. The coverage along each chromosome is indicated by black tickmarks. (A) Patient 9; (B) Patient 14; (C) Patient 15; (D) Patient 16; (E) Patient 30; (F) Patient 42; (G) Patient 43; (H) Patient 59. Several LOHs are recurrent (right arm of chromosome 3, left arm of chromosome 5, and chromosome 1). Please note: data in *Figure 3—source data 1* also applies to this figure.

The following source data is available for figure 4:

**Source data 1**. Persistent LOH regions LOH map.

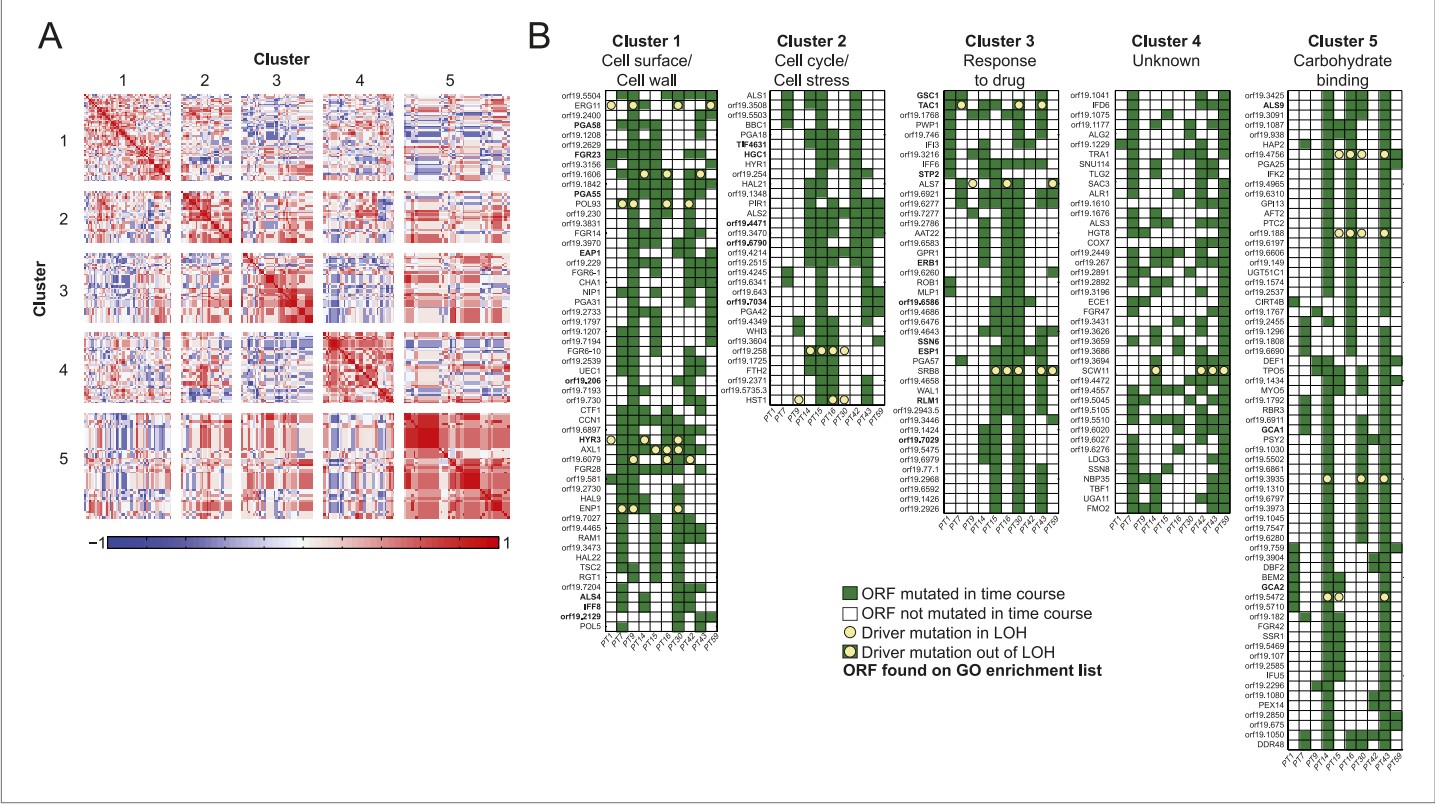

**Figure 5**. Co-occurrence of nonsynonymous substitutions across isolates reveals functional clusters. (**A**) For each of the recurrently mutated 240 genes (genes in which nonsynonymous persistent SNPs appear in more than three patients and are not within an LOH region), we constructed a patient-by-gene binary vector. We clustered the resulting patient-by-gene matrix using NMF clustering to reveal five coherent clusters (correlation matrix of the clusters left; red: positive correlation; blue: negative correlation; white: no correlation). (**B**) Co-occurrence clusters. For the genes in each cluster (rows), shown are their mutated occurrences in each patient (columns); green: gene is persistently mutated in patient, white: no persistent mutation, yellow circle: driver mutation. Functional enrichment of clusters was revealed using gene ontology, and genes matching the enriched cluster function are bolded. We have overlaid recurrent driver mutations (e.g., G/T > A/A) (n = 17) occurring outside of LOH regions (yellow circle, green box) and inside LOH regions (yellow circle, white box).

The following source data and figure supplement are available for figure 5:

**Source data 1**. (A) Recurrence lists and clusters. 1 All Pers NS Genes.

**Figure supplement 1**. Co-occurrence of nonsynonymous SNPs occurring in conjunction with a shift in MIC.

irreversibility of LOH events in the absence of mating highlights the reversible nature (instability) of aneuploidy chromosomes.

While we cannot definitively infer an ordering of events from our singly sampled isolates, we hypothesize that aneuploidy could contribute to the evolution of LOH by increasing the likelihood of its occurrence. For example, in 4 of the 6 patients with a Chromosome 5 LOH, the isolate with an LOH event also harbors a Chromosome 5 trisomy or is preceded by an isolate with a Chromosome 5 trisomy. Thus, the additional copy may increase the likelihood of an LOH event on that chromosome. In three of these cases, *ERG11*, located on the region of Chromosome 5 with LOH, was mutated. Additionally, isolates in 2 of the 7 patients with a Chromosome 3 LOH were trisomic for this chromosome.

## Persistent SNPs in 240 recurrently polymorphic genes identify targets likely associated with drug resistance and host adaptation

We identified persistent nonsynonymous coding SNPs within 1470 genes outside LOH tracts, 167 of them harboring 336 driver-like polymorphisms (*Figure 5—source data 1B*). These again include *ERG11* in patients 9, 14, 30, and 59 and *TAC1* in patients 1, 7, 14, 15, 30 and 43 (*Figure 5*). Applying

the recurrence filter (i.e., persistent nonsynonymous SNPs that appeared in the same ORFs in three or more patient series), we identified 240 polymorphic genes that are more likely to have contributed to adaptation (*Figure 5—source data 1A*). This number of genes is higher than expected by chance (empirical p < 10⁻⁴ based on a Poisson model of background mutation, 'Materials and methods'). Though the coding sequence for these 240 recurrent genes is longer than average (2.21 ± 1.53 kb vs 1.83 ± 1.29 kb for non-recurrent persistent genes, p < 3.68 × 10⁻⁵, *t*-test), and thus a larger target for mutation, our simulation accounts for gene length. Notably, 17 persistent recurrently polymorphic genes also had driver-like polymorphisms, eight of which were also homozygosed in an LOH tract in at least one patient series (*Figure 5*, *Figure 5—source data 1A,B,C*). Finally, polymorphisms in 166 of the 240 genes appeared together with an increase in MIC and are thus stronger candidates for making a significant functional contribution to resistance (*Figure 5—figure supplement 1*, *Figure 5—source data 1D*, empirical p < 10⁻⁵ based a binomial model, 'Materials and methods').

The set of 240 recurrently mutated genes was enriched for fungal-type cell wall (18 genes, p < 0.0012) and cell surface genes (24 genes, p < 0.00012), including several members in each of three cell wall gene families important for biofilm formation and virulence (*Hoyer et al., 2008*): the Hyr/Iff proteins (*HYR1* and *3*, *IFF8* and *6*), the *ALS* adhesins (*ALS1-4,7,9*), and the *PGA-30*-like proteins (seven genes) (*Figure 5—source data 1A*). All three families are specifically expanded in the genomes of pathogenic *Candida* species (*Butler et al., 2009*). In addition, seven members of the *FGR* genes (*Uhl et al., 2003*), involved in filamentous growth and specifically expanded in *C. albicans* (*Butler et al., 2009*), are also among the 240 genes (*Figure 5—source data 1A*).

The most recurrently mutated gene outside of an LOH region was *AXL1* that encodes a putative endoprotease, whose transcript is upregulated in an RHE model of oral candidiasis and in clinical isolates from HIV+ patients with oral candidiasis (*Zakikhany et al., 2007*). The gene is persistently mutated in eight series, (three of which were driver mutations), followed by ten genes mutated in seven series (*Figure 5—source data 1A*). *ERG11*, which encodes the drug target of fluconazole, was affected in 70% (7/10) of the patient series with persistent SNPs in four series (Patients 9, 14, 30, and 59) and mutations in the LOH events in three series (Patients 1, 15, and 43) (*Figure 5—source data 1A*). Likewise *HYR3*, a known virulence gene, was persistently mutated in nine of the patients, three of which occurred in LOH tracts, including one in which a new allele was homozygosed (*Figure 5*, *Figure 5—source data 1A,B,C*). More generally, 171 of the 240 genes were also mutated in an LOH tract in at least one additional patient (15/171 in three or more additional patients and 34/171 in two additional patients).

Next, we partitioned the 240 recurrently mutated genes into 5 'co-occurrence clusters' based on the correlation in their mutation occurrence patterns (*Figure 5*, *Figure 5—source data 1A*). These correlations are significantly higher than expected in a null model (p < 5.2 × 10⁻¹⁸², permutation test, 'Materials and methods'). The characterized genes in most of the clusters have coherent functions. Cluster 1 is enriched for cell wall and cell surface genes, Cluster 2 for cell cycle and stress genes, Cluster 3 for genes involved in drug response, and Cluster 5 for carbohydrate binding (*Figure 5*, *Figure 5—source data 1A*). Most of the genes in these clusters are not well characterized and represent new candidates involved drug resistance and adaptation to the host environment. The full list of genes and descriptions is given in *Figure 5—source data 1A*.

## Changes in virulence phenotypes in evolved drug-resistant isolates

To explore the possibility that some of the mutations reflect adaptation to other factors besides drug, we next measured phenotypes associated to virulence and interaction with the host ('Materials and methods'). Adhesion, filamentation, and virulence in a *C. elegans* model of infection (*Jain et al., 2013*) were measured for a large panel of isolates (*Figure 6*, *Figure 6—figure supplement 1*, *Figure 6—source data 1*). Additionally, we measured competitive fitness in standard tissue culture medium (RPMI) with and without drug in vitro (*Figure 7*).

We found substantial variation in many of these phenotypes between isolates in the same series (*Figure 6* and *Figure 7*), supporting the notion that the isolates are samples from a broad range of genetic variants within clonal (single infection) populations. In general, increased fitness in vitro (in the absence of drug) correlated with an increase in traits associated with virulence (adhesion, filamentation, and virulence in nematode). For example, the later isolates in the series from patients 30 and 43 had increased fitness and higher virulence by all three measures (*Figure 6A,B*, *Figure 7*); whereas, a decrease in fitness in isolate 13 of patient 1 was accompanied by a decrease in virulence (*Figures 6C* and

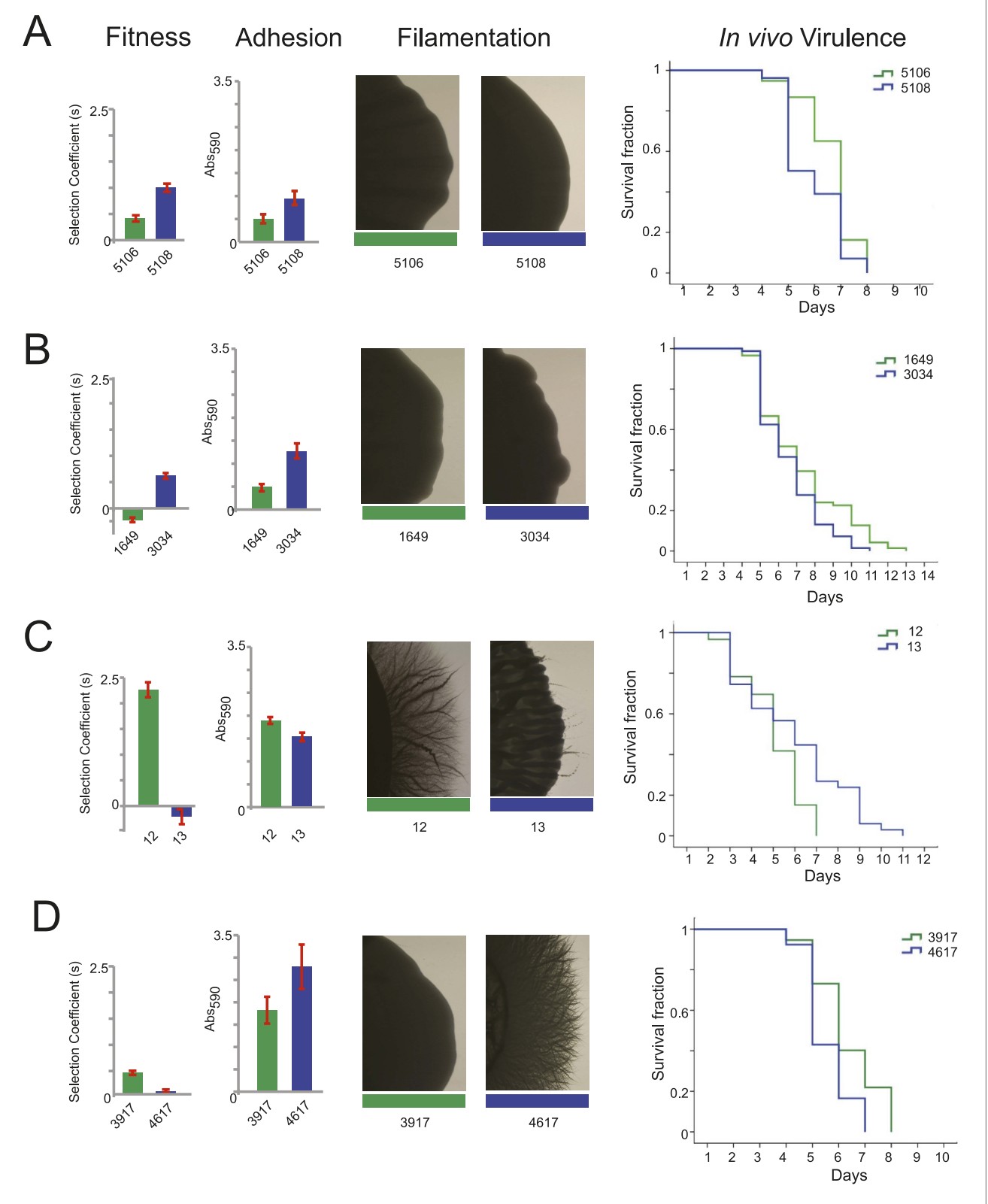

**Figure 6**. Filamentation, adhesion and virulence increase concurrently with fitness. For each pair of consecutive isolates (green preceding blue), shown are the fitness, adhesion, filamentation, and virulence in a worm model of infection (each described in 'Materials and methods'). A subset of fitness values are duplicated from **Figure 7A**, with selection coefficient (s) shown on the Y-axis. A subset of adhesion values are plotted from

*Figure 6. Continued on next page*

*Figure 6. Continued*

*Figure 6—source data 1*, with Abs590 nm on the Y-axis. A subset of images showing filamentation on spider media are shown, with the full set found in *Figure 6—figure supplement 1*. For virulence, shown are Kaplan–Meier plots of survival rates from *C.elegans* infection with the specified *C. albicans* isolates ('Materials and methods'). For each isolate pair, significant changes in virulence were observed between the two isolates (in all cases, p < 0.001, log-rank test), with three of the four evolved isolates being more virulent than their corresponding progenitor. (**A**) Patient 30 isolates 5106 and 5108; (**B**) Patient 43 isolates 1649 and 3034; (**C**) Patient 1 isolates 12 and 13; (**D**) Patient 59 isolates 3917 and 4617.

The following source data and figure supplements are available for figure 6:

**Source data 1**. Adhesion values for the majority of isolates.

**Figure supplement 1**. Filamentation increases in many patient series.

*Figure 7*). A notable exception was patient 59, where fitness in vitro decreased while virulence phenotypes increased in a later isolate (*Figures 6D* and *Figure 7*). This is consistent with the observations of Noble and co-workers that in vitro fitness is not always a reflection of virulence (*Noble et al., 2010*).

Initially in a series, drug resistance (MIC) and in vitro fitness (in the absence of drug) were inversely related, suggesting that these are competing selective pressures. When MIC increases first appeared, they were usually accompanied by a *decrease* in fitness in the absence of drug (Patient 1, isolates 2, 13, and 16, Patients 9, 14, 15 16, and 59, *Figure 7A*). Consistent with the elevated MIC, these isolates exhibited *increased* relative fitness in the presence of the drug (*Figure 7B*). This is also consistent with a recent study (*Sasse et al., 2012*) showing that resistance conferred in *C. albicans* by gain-of-function mutations in the transcription factors Mrr1, Tac1, and Upc2 is associated with reduced fitness under non-selective conditions in vitro as well as in vivo during colonization of a mammalian host. Consistent with subsequent selection of strains with compensatory variations, isolates from later time points were often more fit than those from earlier time points (measured in vitro, in the absence of drug) without further changes in MIC (e.g., patient 1, isolates 5-7, isolate 14, *Figure 7A*), with notable exceptions (e.g., isolates 8 and 11). This general trend is consistent with previous studies in bacteria (*Bjorkman and Andersson, 2000*); (*Gagneux et al., 2006*) and in a single documented case in *C. glabrata* (*Singh-Babak et al., 2012*), suggesting that compensatory mutations may subsequently arise to offset the major fitness cost of mutations conferring drug resistance. Nevertheless, substantial additional sampling will be required per time point to fully interpret such patterns.

In this context, it appears that aneuploidies (*Figure 7A*, green), while largely transient, may be an important intermediate giving rise to more stable adaptive genotypes in some cases, as was recently demonstrated in budding yeast adapting to a stressful environment in vitro (*Yona et al., 2012*). For example, in Patient 1 isolate 13, an increase in MIC and a trisomy of 5 of 8 chromosomes accompanies a large decrease in fitness (in the absence of drug) relative to the preceding isolate 12 (*Figure 7A*) but has increased fitness in the presence of drug (*Figure 7B*). Isolate 14 has a similar MIC phenotype to isolate 13 but is euploid (*Figure 3*) and is much more fit (*Figure 7A*). Consistent with the general negative effect of aneuploidy on fitness (*Tang and Amon, 2013*), the absence of the extra chromosomes resulted in improved overall fitness.

## Candidate mutated genes associated with drug resistance or virulence

The analysis of clinical isolates identified a range of new candidate genes that may affect drug resistance, fitness, and/or virulence. To test the contribution of some of the recurrently identified genes to specific *C. albicans* phenotypes, we profiled all 23 recurrently mutated loci for which a homozygous deletion mutant was available from a deletion strain collection (*Noble et al., 2010*). We measured the MIC of fluconazole and the in vitro fitness in the absence of drug for each of these 23 mutants.

Deletion of one gene (*orf19.4658*) caused a twofold decrease in MIC, whereas the other 22 mutants tested did have no significant effect on MIC (*data not shown*). Deletion mutants are loss-of-function mutations, whereas the previously identified mechanisms of fluconazole resistance are 'gain of function', resulting in the increase in the amount or activity level of Erg11 (*Asai et al., 1999*; *Oliver et al., 2007*) or the efflux of drug transporters (*Coste et al., 2006*; *Dunkel et al., 2008*). Therefore, it is possible that the recurrent non-synonymous coding SNPs in the new loci, we identified in the clinical isolates confer resistance. Alternatively, these loci may not be involved in fluconazole resistance per se and may have a more general role in adaptation to the complex host environment.

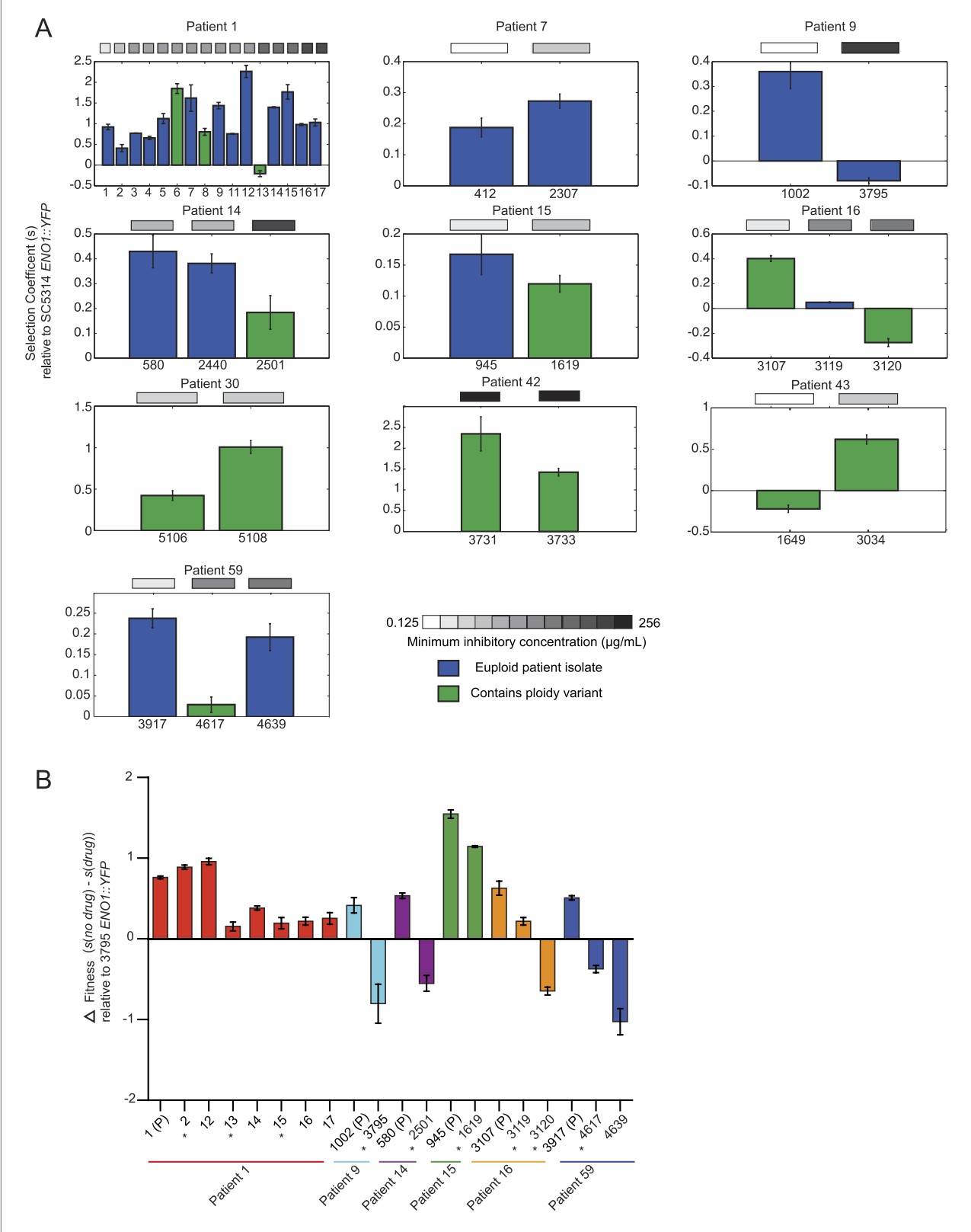

**Figure 7**. Emergence of increased drug resistance often coincides with reduction in fitness in the absence of drug, but an increase in the presence of drug. (**A**) For each patient (panel) shown is the fitness ('Materials and methods') of each strain (Y axis, mean ± STDV), ordered from the progenitor to evolved isolates (left to right, X axis). Fitness is calculated relative to an ENO1::YFP SC5314 reference isolate. The MIC of each strain is shown in the gray

*Figure 7. Continued on next page*

*Figure 7. Continued*

boxes on top (white: low; black: high, color bar at bottom). Green: isolates with aneuploidies; Blue: euploid isolates. (**B**) Shown is the mean difference between fitness in the absence and presence of drug (Y axis, error bars are ± STDV; n ≥ 3) for isolates (X axis) that showed a decrease in fitness (*Figure 7A*) in the absence of drug concomitant with an increase in MIC (asterisks), and flanking isolates in Patient 1 and 59 (ordered from the progenitor to evolved isolates, left to right, X axis). The difference in fitness is calculated as the difference in selection coefficient (s, Y axis) between matching competition experiments in RPMI and those in RPMI with one half the MIC for fluconazole (*Table 1*) for each isolate tested (X axis). Negative values indicate that the strain had higher fitness in the presence of fluconazole vs assays without fluconazole. For each assay, the fluconazole-resistant isolate 4639 *ENO1::YFP* was used as the reference strain.

Consistent with a role in host adaptation, 5 of the 22 deletion mutants reduced in vitro fitness in a culture medium thought to approximate in vivo conditions (*Figure 8*, 'Materials and methods'). Three were significantly more fit than the WT parental strain (SN250, red, *Figure 8*), including *CCN1*, that encodes a G1 cyclin required for hyphal growth maintenance (*Loeb et al., 1999*) and *orf19.4471*, an ortholog of *Saccharomyces cerevisiae VPS64*, which is required for cytoplasm-to-vacuole targeting of proteins (*Bonangelino et al., 2002*), is involved in recycling pheromone receptors (*Kemp and Sprague, 2003*), and is identified as an 'aneuploidy-tolerating mutant' (*Torres et al., 2010*). Among the least fit were cell wall protein genes (*HYR1, HYR3,* and *PIR1*; *De Groot et al., 2003*).

## Discussion

We sequenced the genomes of serial clinical isolates of *C. albicans* and analyzed them by comparing consecutive isolates from one patient to reach novel insights into drug resistance within the human host. This approach allowed us to distinguish (and remove from further analysis) isolates that were non-clonal and to estimate that at least ~30% of the patients (3/10) carried at least one non-clonal strain of *C. albicans.* We used the clonal isolates to identify persistent SNPs, and the different series to identify those persistent SNPs that recurred within the same ORFs, thereby focusing the analysis on a small number of loci where the identified variants are more likely to be adaptive and excluding the substantial background of likely neutral variation that hitchhike along with selective beneficial mutations.

Our study identified substantial genetic diversity in each series, in contrast to the report of only 26 SNPs detected in a single clinical series of *Candida glabrata* isolates that spanned a 10-month period (*Singh-Babak et al., 2012*). Several reasons may account for this difference. First, fluconazole, the antifungal drug used to treat the patients in our series, is fungistatic, such that many cells exposed to the drug arrest their growth but do not die. Thus, the range of diversity in the initial population is not entirely lost. In contrast, the *C. glabrata* study involved exposure to caspofungin, an echinocandin fungicidal drug. Therefore, most cells likely died upon drug exposure and only the rare survivors went on to seed the remaining population. Accordingly, the *C. glabrata* isolates may have been subjected to selection that would have removed much of the initial diversity in the population, whereas in the *C. albicans* series diversity persisted and selection acted mostly to change the relative proportions of different genotypes. Second, *C. albicans* is a highly heterozygous diploid whereas *C. glabrata* is haploid. Mutations can be more readily assimilated in a diploid than in a haploid organism, since deleterious mutations are potentially buffered by a functional version (*Thompson et al., 2006*). Furthermore, because *C. albicans* genomes are initially much more diverse (with tens of thousands of heterozygous SNPs in a given isolate), LOH is a high frequency mechanism available to reveal mutations more readily. Third, *C. albicans* lab isolates likely undergo a stress-induced elevation of mutation and mitotic recombination rates (*Ponder et al., 2005*; *Forche et al., 2011*; *Rosenberg, 2011*), and exposure to a mammalian host results in elevated frequencies of LOH and aneuploidy (*Forche et al. 2009*). Thus, it is possible that *C. albicans* isolates within the human host also undergo elevated levels of LOH and of point mutations to generate a wider range of diversity. Thus, *C. albicans* like *S. cerevisiae* (*Gresham et al., 2008*; *Pavelka et al., 2010*; *Yona et al., 2012*; *Chang et al., 2013*) generates large scale genetic variation as a means of adaptation. This adds another level of variation to the genome and protein diversity (*Santos et al., 2004*; *Selmecki et al., 2010*) that *C. albicans* is able to tolerate.

LOH in several genes important for fluconazole resistance has been reported previously for *ERG11* (*Oliver et al., 2007*), *TAC1* (*Coste et al., 2006*), and *MRR1* (*Schubert et al., 2011*), but the degree to

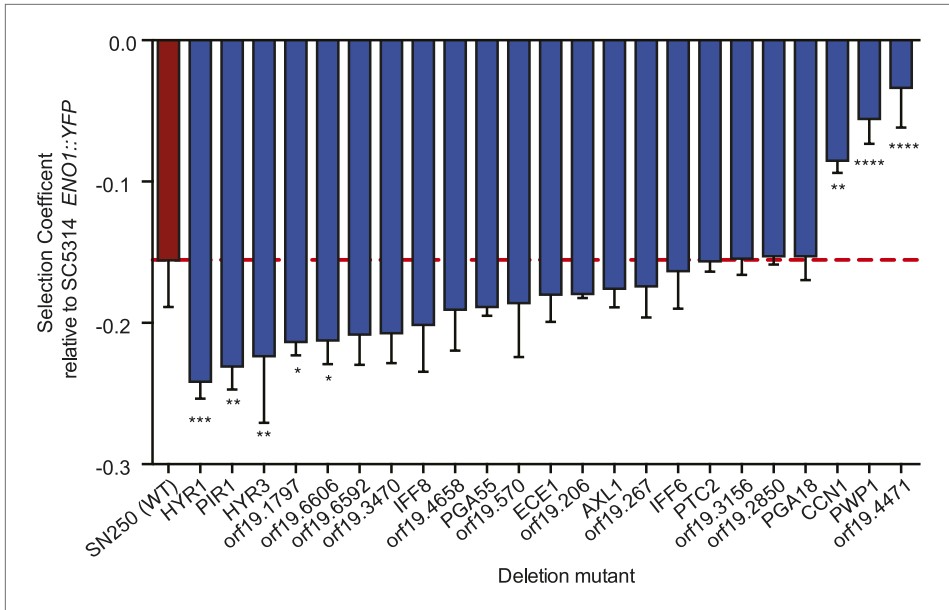

**Figure 8**. Deletion mutants of recurrently mutated genes reveal changes in relative fitness. Shown is the fitness ('Materials and methods') for each deletion mutant strain and the corresponding wild-type strain (Y axis, mean ± STDV). The wild-type parental strain (SN250) is on the far left (red bar and dashed line). Fitness is calculated relative to an ENO1::YFP SC5314 reference isolate. Locus names are given for the mutant isolates (X axis). Asterisks denote statistical significance (* < 0.05, ** < 0.01, *** < 0.001, **** < 0.0001) by one-way ANOVA with Holm–Sidak correction for multiple comparisons.

which LOH is important in clinical infections was not known. In the ten patients studied here, LOH was commonly observed and was associated with changes in MIC. As we detected LOH of mutations in *ERG11* in three patients, it would be of interest to know if the LOH in these known genes was sufficient to increase MIC, or if other genes within the homozygous region make important contributions.

Aneuploidies appeared frequently within the drug-resistant isolates, consistent with previous reports (*Selmecki et al., 2006*). Unlike LOH events, aneuploidies were often transient and not consistently correlated to increases in drug resistance. Perhaps these aneuploidies provide a mechanism akin to genetic assimilation ('phenotype precedes genotype'), in which cells are provided with a phenotypic mechanism that facilitates survival until a more stable and/or less costly mechanism is attained. In this case, the 'phenotypic' mechanism would be genetic but unstable—the acquisition of one or more extra chromosomes. Nevertheless, aneuploidies may cause increased frequencies of LOH events through whole chromosome loss, as well as by increasing the likelihood of recombination events. A transient role for aneuploidy is consistent with recent findings from in vitro evolution studies in *S. cerevisiae* (*Yona et al., 2012*) in which a transient aneuploidy was responsible for fitness at elevated temperature, but was eventually replaced by a more stable mutation.

In addition, a substantial number of persistent and recurrent SNPs, and clusters of co-occurring SNPs, implicate a broad range of pathways and functions that likely provide some growth advantage in the presence of the complex selective pressures found in the host. In particular, there was strong enrichment for cell wall gene families thought to be critical determinants of the transition from commensalism to pathogenesis (*Gow and Hube, 2012*). The genes in several of these families (e.g., *ALS1-4,7,9* and *HYR/IFF* genes) frequently contain intragenic tandem repeats. Variation in intragenic repeat number modulates phenotypic diversity in adhesion and biofilm formation (*Verstrepen et al., 2005*). This functional diversity of cell surface antigens has been proposed to allow rapid adaptation to the environment as well as evasion of the host immune system in fungi and other pathogens (*Gemayel et al., 2010*). Notably, the cell wall deletion mutants (*HYR1*, *HYR3*, and *PIR1*) were among the least fit in vitro (*Figure 8*).

Indeed, many of the isolates evolved additional phenotypes, including changes in in vitro fitness, filamentation, adhesion, and in vivo virulence, and the data presented here points to candidate

genes that underlie some of these evolved traits. For example, the evolved isolate 4617 in patient 59 had a dramatic increase in filamentation relative to the progenitor, which was concomitant with the appearance of persistent SNPs in genes associated with filamentous growth: *CHO1*, *MNN2*, and 7 different *FGR* (filamentous growth regulator) genes (*Uhl et al., 2003*).

The evolution of drug resistance in *C. albicans* has many parallels with the somatic evolution of cancer cells undergoing chemotherapy or treated with specific inhibitors. These include variation on a background of clonal descent, lack of sexual recombination, acquisition of drug resistance, tolerance of aneuploidy and genome plasticity, and increased mutation and mitotic recombination rates under stress. Indeed, several recent studies have shown a similar spectrum of genetic alterations to those observed here during the somatic evolution of cancers in patients undergoing chemotherapy (*Podlaha et al., 2012*; *Landau et al., 2013*) or treated with specific inhibitors (*Ding et al., 2012*) to those observed here.

Finally, our data and analyses provide a rich and novel resource for *Candida* researchers and a host of candidate genes for further functional studies. While our analysis focused on recurrent SNPs in ORFs, we nonetheless cataloged the many genetic alterations found in intergenic regions (*Figure 2—source data 1*), some of which could affect gene regulation. It will be especially interesting to analyze the similarities and differences in additional *C. albicans* genome sequences that are likely to become available in the near future. Our results suggest there may be complex population dynamics during the transition from commensal to pathogen and across the course of treatment. As sequencing capacity continues to grow, it will be especially interesting to more fully sample this population-level diversity during longitudinal collection to better understand these dynamics. In particular, it will be interesting to determine the degree to which specific mutations recur in different isolates prior to and after the acquisition of drug resistance.

## Materials and methods

### Isolates

Isolates were obtained from HIV-infected patients with oropharyngeal candidiasis, as previously described (*White, 1997a*; *Perea et al., 2001*). The patients were not on azole antifungal treatment at time of enrollment; subsequent samples were collected during recurrence of infection. Isolates were colony purified at collection and represent a single clone. The isolates are detailed in *Table 1*.

### Drug susceptibility

Minimal inhibitory concentrations (MIC) were determined for each strain (clinical and mutant) using fluconazole E-test strips (0.016–256 µg/ml, bioMérieux, Durham, NC) on RPMI 1640-agar plates (Remel, Lenexa, KS). Overnight YPD cultures were diluted in sterile 0.85% NaCl to an OD600 of 0.01 and 250 µl was plated using beads. After a 30-min pre-incubation, 2–3 E-test strips were applied and plates were incubated at 35°C for 48 hr. The susceptibility endpoint was read at the first growth-inhibition ellipse, and the median value is reported here.

### Illumina sequencing

Genomic DNA was prepared from different clinical time courses via a Qiagen Maxiprep kit (Qiagen, Valencia, CA) and sequenced using 101 base paired-end Illumina sequencing (*Mardis, 2008*). Library preparation included an eight base barcode (*Grabherr et al., 2011*); 43 samples from 11 patients were sequenced. All reads were mapped to the SC5314 reference genome (Candida Genome Database Assembly 21, gff downloaded on 4 January 2010) using the BWA alignment tool (version 0.5.9) (*Li and Durbin, 2009*). To minimize false positive SNP calls near insertion/deletion (indel) events, poorly aligning regions were identified and realigned using the GATK RealignerTargetCreator and IndelRealigner (GATK version 1.4-14, [version 1.4-14]) (*McKenna et al., 2010*). Coverage for each strain is reported in *Table 1*. Coverage was defined as the total number of bases with BWA mapping quality greater than 10 divided by the total number of sites in the nuclear genome. Isolate 4380 aligned poorly to the SC5314 genome; however, this sequence aligned at high identity to the *C. dubliniensis* genome and was therefore removed from further analysis. These data can be accessed from a genomics portal hosted by the Broad Institute at: http://www.broadinstitute.org/pubs/candidadrugresistance/. Reads are deposited for access to the NCBI SRA under project accession number PRJNA257929.

## SNP calling

SNPs were identified using Unified Genotyper (GATK version 1.4.14) (*McKenna et al., 2010*), using read alignments to the SC5314 reference sequence. Unreliable SNPs were identified using the GATK Variant Filtration module, with the version 3 best practice recommended annotation filters (QD < 2.0, MQ < 40.0, FS > 60.0, HaplotypeScore >13.0, MQRankSum < −12.5, ReadPosRankSum < −8.0) except that the HaplotypeScore was also filtered if greater than two standard deviations above the mean of all HaplotypeScore values. The combined list of SNP positions across all strains was used to evaluate those matching the reference allele; by emitting all sites using Unified Genotyper, high quality reference matches were identified as positions with quality of 30 or greater, with positions with extremes of read depth (top or bottom 0.5% quantile) eliminated. A matrix of all strains by all positions was created from the SNP calls, with reference calls added where identified. Non-clonal isolates (see below) were removed from further analysis.

## Sequenom iPLEX genotyping assay

We chose 1973 genetic locus X strain combination (523 unique sites across nine patients) for iPLEX genotyping as either (1) persistent within their time course (605 sites), (2) background mutations (1263 sites), or (3) transient mutations (105 sites). All selected loci were at least 150 bp away from any other SNP in either direction to avoid ambiguous iPlex calls. Sites producing multiple iPlex results were eliminated from further consideration. 1,853 predictions were confirmed as correct by Sequenom genotyping and 120 were discordant (*Table 2*), to a calling accuracy of 93.9%.

## Determination of relatedness to determine clonality

We investigated the phylogenetic relationship of all strains using SNP calls to determine relatedness between strains; positions with missing data in 10% or more of strains were eliminated, resulting in a total of 201,793 parsimony informative positions. A distance based tree was estimated using maximum parsimony with PAUP* (4.0) (*Swofford, 2002*); a step matrix was used to estimate the distance between homozygous and heterozygous positions, where each of the homozygotes is two steps apart from each other and one step from the heterozygote. SNP positions were resampled using 1000 bootstrap replicates, and the phylogeny re-estimated to test the branch support. We define isolates with a branch distance of greater than 20,000 as non-clonal.

## Copy-number determination

For each strain, we calculated a per-locus depth-of-coverage using GATK (*McKenna et al., 2010*), with a minimal mapping quality of 10. The number of reads aligning to each 5 kb window across the nuclear genome was calculated and then normalized to the genome median. Each bin was then multiplied to the ploidy for the majority of the genome as determined by a FACS assay (below). We then applied a sliding window across each bin, defining a potential CNV if 70% of 10 consecutive bins had a normalized count >2.5×. Regional amplifications are identified if >15% of the chromosome is identified as having a CNV. Boundaries were confirmed by visual inspection in the Integrative Genome Viewer (*Robinson et al., 2011*).

## High-resolution ploidy analysis by flow cytometry

*C. albicans* cultures were grown to log phase. 200 µl of culture was centrifuged in a round bottom microtiter plate, and pellets were resuspended in 20 µl of 50 mM Tris pH8/50 mM EDTA (50/50 TE). 180 µl of 95% ethanol was added and suspensions were stored overnight at 4°C. Cells were centrifuged and pellets washed twice with 200 µl of 50/50 TE, then resuspended in 50 µl of RNAse A at 1 mg/ml in 50/50 TE and incubated 1 hr at 37°C. Cells were centrifuged and pellets resuspended in 50 µl of Proteinase K at 5 mg/ml in 50/50 TE for 30 min at 37°C. Cells were washed in 50/50 TE and pellets resuspended in 50 µl of a 1:85 dilution SYBR Green I (Invitrogen, Carlsbad, CA) in 50/50 TE and incubated overnight in the dark at 4°C. Cells were centrifuged and pellets were resuspended in 700 µl 50/50 TE and read on a FACS caliber flow cytometer (BD Biosciences, San Jose, CA). Flow data were fitted with a multi-Gaussian cell cycle model to produce estimates for whole genome ploidy.

## LOH determination

For each time course, we assembled the high quality SNPs (post-filtering, above) from multi-sample calling into the columns of a matrix, ordered by genome position, with the isolates in rows, ordered temporally. The genetic state of each locus in each sample was coded to distinguish loci homozygous

for the haploid reference (−1), heterozygous SNPs (0), and homozygous SNPs for the non-reference state (1). We then applied a sliding window method across each chromosome, only looking at sites in which a SNP call was made in at least one isolate. An LOH event was defined as occurring if (1) at least one isolate had a heterozygosity content >40%, and (2) at least one other isolate had a heterozygosity content <5%. Window sizes were of length 500. Boundaries were trimmed such that if a window terminated in a heterozygous site in the isolate for which the LOH occurred, it was trimmed back until it was homozygous. If two 500+ windows were within 7 KB of each, the region was assessed to determine if the event was actually one event and merged if the heterozygous sites in the inter-window space had homozygosed. If two isolates had LOHs that overlapped but did not have precisely identical boundaries, the LOH regions were combined such that the LOH interval for both isolates was the same. All LOH regions were confirmed by visual inspection and are listed in *Figure 3* and *Figure 4— source data 1*.

## Classification of SNPs

For each time course, each SNP was classified for its position in the genome (*Figure 2—source data 1*). If the SNP fell within an ORF, the reference and altered amino acids were reported. If the SNP fell outside of an ORF, the distance to the closest flanking ORF(s) was reported, as well as the SNP's orientation with respect to these ORFs. SNP genotypes that are common to all isolates (including the 'progenitor') were classified as background mutations. Genotypes not present in the progenitor or evolved strain, but that occur in one or more intermediate strain, are classified as transient. Finally, genotypes that occur after the progenitor, and persist through the terminally evolved time point, are classified as persistent.

To determine if the number of persistent non-synonymous SNPs (nsSNPs) occurring in conjunction with changes in MIC was greater than expected, we developed a simple model to simulate the occurrence of nsSNPs outside of LOH regions at each time point. For each time point (i), a random variable $X_i \sim \text{Pois}(\lambda_i)$ was assigned, where $\lambda_i$ represents the Poisson parameter for each time point:

$\lambda_i = m/T \times t_i$;
m = 1471, the number of ORFs with persistent nsSNP;
T = 23, the number of time points;
$t_i$ = the length of time (days) for time point (i) divided by the mean length of time.

The number of persistent nsSNP-containing ORFs for each of the 14 time points associated with a change in MIC was summed, and this was repeated 100,000 times to build a probability distribution, where p (observing x mutated ORFs) was determined by dividing the number of successes for each bin by the number of trials.

To determine the probability of observing x recurrent, persistently mutated ORFs outside of LOH regions, we developed an additional stochastic simulation model. For each patient series (i), at each non-LOH ORF (j), a random variable, $X_{ij} \sim B(n, p_{ij})$ was assigned, where n represents the number of trials, 1, and p represents the probability of a SNP occurring in that ORF:

p = $m_i/M_i \times h_j$;
$m_i$ = number of ORFs with persistent nsSNPs found outside of LOH events for patient series (i);
$M_i$ = number of ORFs outside of LOH events for patient series (i);
$h_j$ = log normalized ORF length divided by mean lognormalized ORF length for ORF (j).

This was repeated 10,000 times to build a distribution, and p (observing x recurrent nsSNPs) was determined by dividing the number of successes in each bin by the cumulative number of trials.

## Analysis of co-occurring mutations

For co-occurrence analysis we focused only SNPs that (1) had persistent nonsynonymous coding SNPs that did not occur in LOH regions and (2) recurred in three or more time courses. We generated for each such gene a binary patient vector, and we used NMF clustering (*Brunet et al., 2004*) to identify the optimal number of clusters, based on local maximas. This was accomplished using the 'NMFConsensus' module (version 5) in GenePattern (*Reich et al., 2006*). To determine the most appropriate number of clusters, k was selected such that it was the smallest value for which the cophenetic correlation begins decreasing. We then tested each of the co-occurrence gene clusters for functional enrichment (below). To determine if the degree of co-occurrence would have arisen by chance, we ran 1000 iterations of 1 million edge-pair swaps from the original binary matrix,

calculating a Pearson correlation matrix for each of the 1000 iterations. We compared the distribution of Pearson correlations on the real and permuted vectors using a two-sample Kolmogorov–Smirnov (KS) test and Wilcoxon Rank Sum test.

## Functional enrichment

We calculated the overlap of each co-occurring cluster with Gene Ontology gene sets using the Gene Ontology toolset from the Candida Genome Database (*Arnaud et al., 2009*; *Inglis et al., 2012*, *2013*). Bonferroni adjusted p-values as well as the False Discovery Rate are reported (*Figure 5—source data 1A*).

## Competition assay to assess fitness

We measured the relative fitness of the progenitor and evolved lines in RPMI Cell Culture medium (Gibco, Grand Island, NY), competing them against a reference strain (SC5314), expressing *ENO1::YFP*. Isolates stored at −80°C were revived on rich media petri plates and then grown overnight in 3-ml cultures of minimal media in a roller drum at 35°C. An aliquot of cells in each culture was removed, sonicated in a Branson 450 sonifier, and the concentration of cells was determined using a Cellometer M10 (Nexcelom, Lawrence, MA). The reference strain and experimental competitors were added to fresh RPMI medium in a 1:1 ratio and a final cell concentration of $1 \times 10^7$ cells/ml. The cultures were grown for 24 hr in a roller drum at 35°C. Cells were then counted as above, and $3 \times 10^6$ cells were transferred to fresh RPMI medium grown for 24 hr in a roller drum at 35°C (transfer cycle 1). This procedure was repeated (transfer cycle 2). This protocol represents 5–10 generations of growth, depending on the strain genotype. The ratio of the two competitors was quantified at the initial and final time points by flow cytometry (Accuri, San Jose, CA). 3 to 6 independent replicates for each fitness measurement were performed. The selective advantage, *s*, or disadvantage of the evolved population was calculated as previously described (*Thompson et al., 2006*), where *E* and *R* are the numbers of evolved and reference cells in the final (*f*) and initial (*i*) populations, and *T* is the number of generations that reference cells have proliferated during the competition.

## Competition assay to assess fitness with and without fluconazole

Fitness assays were performed as described above except that the reference strain used was a derivative of the drug-resistant isolate 4639 from patient 59 (*Table 1*) expressing *ENO1::YFP* and competition experiments were performed in RPMI and in RPMI with one half the MIC for fluconazole (*Table 1*) initiated from replicate 1:1 mixtures of the same population of cells for each isolate tested.

In order to quantify fitness in the presence of fluconazole, we constructed a derivative of the fluconazole-resistant (128 μg/ml) isolate 3795 from patient 9 (*Table 1*) that expresses *ENO1::YFP* to use as the reference for competitive fitness assays in the presence and absence of fluconazole. This strain was chosen since it was a euploid strain with the highest MIC. There were two strains from patient 42 (3731 and 3733) with a higher MIC but these strains are aneuploid and thus the potential loss of additional chromosomes during the course of the competition could alter fitness and confound our results. The addition of the YFP marker reduced the fitness of the strain relative to the unaltered one and slightly reduced the fluconazole MIC as measured by the E-strip test. Fitness assays were performed as described above competition experiments except they were performed in RPMI and in RPMI with one half the MIC for fluconazole (*Table 1*) initiated from replicate 1:1 mixtures of the same population of cells for each isolate tested.

## *C. elegans* survival assay

A *C. elegans* survival assay was performed as previously described (*Jain et al., 2009*). Briefly, *Escherichia coli* OP50 and the different *C. albicans* clinical isolates were grown overnight respectively in LB at 37°C and YPD at 30°C. *E. coli* was then centrifuged and resuspended to a final concentration of 200 mg/ml, while *C. albicans* isolates were diluted with sterile water to $OD_{600} = 3$. Small petri dishes (3.5 cm) containing NGM agar were spotted with a mixture of 10 μl streptomycin (stock solution 50 mg/ml), 2.5 μl of *E. coli*, 0.5 μl of *C. albicans*, and 7 μl of sterile water. The plates were incubated overnight at 25°C and 20 young synchronized N2 *C. elegans* adults were transferred on the spotted plates. Synchronous populations of adult worms were obtained by plating eggs on NGM plates seeded with *E. coli* OP50 at 20°C for 2–3 days. In this time frame, the eggs hatch and the larvae reach young adulthood. The survival assay was carried at 20°C, and worms were scored

daily by gentle prodding with a platinum wire; dead worms were discarded while live ones were transferred to seeded plates grown overnight at 25°C. Worms accidentally killed while transferring or found dead on the edges of the plates were censored. Statistical analysis was performed using SPSS software; survival curves were obtained using the Kaplan–Meier method and p-values by using the log-rank test.

## Filamentation assay

Overnight cultures grown in YPD at 30°C were normalized to $OD_{600} = 1$ with sterile water and spotted on Spider agar media (1% mannitol, 1% Difco nutrient broth, 0.2% $K_2HPO_4$). Plates were incubated at 37°C and colonies were photographed 3, 7, and 10 days post spotting. As a negative control for filamentation cph1/cph1 efg1/efg1 (*Lo et al., 1997*) double mutant strain was used.

## In vitro adhesion assay

The in vitro adhesion assay was performed as previously described for *S. cerevisiae* (*Reynolds and Fink, 2001*). Briefly, cultures were grown in Synthetic Complete (SC) media + 0.15% glucose at 30°C overnight. Cells were then centrifuged at maximal speed and resuspended to $OD_{600} = 0.5$ in fresh media. 200 ml of each culture were dispensed into 8 wells of a flat bottom 96-well plate and incubated at 37°C for 4 hr. The content of the plate was then decanted and 50 ml of crystal violet added to each well. After 45 min of incubation at room temperature, the content of the plate was decanted and the plate was rinsed ten times in DI water by alternate submerging and decanting. 200 ml of 75% methanol was added to each well and absorbance was measured after 30 min at $OD_{590}$. An edt1/edt1 knockout mutant (*Zakikhany et al., 2007*) was used as a negative control for adhesion.

## Additional information

### Competing interests

AR: Senior editor, *eLife*. The other authors declare that no competing interests exist.

### Funding

| Funder | Grant reference number | Author |
| --- | --- | --- |
| National Science Foundation | Graduate Student Fellowship | Jason M Funt |
| Howard Hughes Medical Institute | Full investigator | Aviv Regev |
| Helen Hay Whitney Foundation | Postdoctoral Fellowship | Christopher B Ford |
| National Institutes of Health | 8DP1CA174427 | Aviv Regev |
| National Institutes of Health | 2R01CA119176-01 | Aviv Regev |
| Alfred P. Sloan Foundation | | Aviv Regev |

The funders had no role in study design, data collection and interpretation, or the decision to submit the work for publication.

### Author contributions

CBF, JMF, DAT, Conception and design, Acquisition of data, Analysis and interpretation of data, Drafting or revising the article; DA, LI, Acquisition of data, Analysis and interpretation of data; CG, CC, Conception and design, Acquisition of data, Analysis and interpretation of data; DAM, Conception and design, Analysis and interpretation of data, Contributed unpublished essential data or reagents; TD, BL, Conception and design, Acquisition of data, Contributed unpublished essential data or reagents; TCW, Provided many of the fungal samples used in the study, Provided essential domain expertise necessary to interpret results; RPR, JB, AR, Conception and design, Analysis and interpretation of data, Drafting or revising the article

### Author ORCIDs

Toni Delorey, http://orcid.org/0000-0001-6614-3803
Christina Cuomo, http://orcid.org/0000-0002-5778-960X

## Additional files

### Major dataset

The following dataset was generated:

| Author(s) | Year | Dataset title | Dataset ID and/or URL | Database, license, and accessibility information |
|---|---|---|---|---|
| Cuomo C and Ford C | 2014 | The evolution of gradual acquisition of drug resistance in clinical isolates of *Candida albicans* | PRJNA257929; http://www.ncbi.nlm.nih.gov/sra | Publicly available at NCBI Sequence Read Archive. |

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
