## [Decision Letter]

Thank you for sending your work entitled “The mutational landscape of gradual
acquisition of drug resistance in clinical isolates of *Candida
albicans*” for consideration at *eLife*. Your article has
been evaluated by a Senior editor and 3 reviewers, one of whom is a member of our Board
of Reviewing Editors. The manuscript was considered appropriate for
*eLife* but some serious concerns need to be addressed successfully
before a final decision can be reached.

The following individuals responsible for the peer review of your submission want to
reveal their identity: Manolis Dermitzakis (Reviewing editor), Jacques Fellay (peer
reviewer), and Daniel Wilson (peer reviewer).

The Reviewing editor and the other reviewers discussed their comments before we reached
this decision, and the Reviewing editor has assembled the following comments to help you
prepare a revised submission.

The manuscript by Funt and colleagues provide an in-depth analysis of whole genome
sequence of *C. albicans* progressively in HIV patients to infer
selective events and describe patterns of mutations, and those that survive and likely
have an impact on resistance. Overall, the study is large and the data collected is very
important and relevant with a degree of phenotypic assessment paired with the sequence
data. The experimental and sequence analysis methodologies seem to be state of the
art.

1) One major concern arises from consideration of all sections of the manuscript. There
is very little statistical modeling to assess how likely these combinations of certain
mutational patterns are due to real selective events due to specific phenotypic
consequences or just by chance. Most of the manuscript is presented in this style, and
it is hard to know which signals are truly significant. We provide two of the many
examples in the manuscript to illustrate this point:

In the Results section, the authors state: “We consider… two to three
isolates”. While this is a reasonable argument, without proper statistics it is
not possible to evaluate it. Perhaps some degree of linkage disequilibrium would be
useful to consider.

In the Results section, the authors also mention: “The recurring appearance of
LOH events that coincide with changes in MIC suggests that they have been positively
selected…” This is a “coincidence”, as the authors say, but
is it significant?

2) The last part of the Results describes some validation experiments where the specific
mutations are brought to an experimental setting with deletion mutants. This is an
experiment that was necessary but surprisingly it was not given the proper weight (it
was only a small fraction of the total text), plus it was not as conclusive as one would
like regarding the MIC phenotype.

3) Considering the central role in the paper of the various classes of mutations called
from the sequencing reads, I think that one technical issue in particular deserves
further exploration and/or discussion. The SNP calling accuracy was found to be 87.3% on
average (using a Sequenome iPLEX genotyping assay; Results, “Sequenome iPLEX
genotyping assay”, and Table 2). There
was however an important inter-sample variability, with very poor results observed in
half of the tested series (<65% calling accuracy). This is worrying. Could this be
due to intra-patient diversity, with different *Candida* subpopulations
being sampled for the two assays? Is there any other potential explanation? If not,
these inconsistencies should be listed as an important limitation of the study.

4) The paper is presented as a study of the within-host mutations that occur and are
selected for in the *Candida albicans* genome between longitudinally
sampled genomes before and after the introduction of antibiotic treatment. However, the
data suggest that the variants that distinguish *C. albicans* genomes at
different time points did not arise during the sampling window but were largely, maybe
entirely, pre-existing (standing variation). This view is based on: (i) the very
considerable genetic diversity (thousands of SNP differences) between isolates of the
same strain within the same host, sampled less than a few months apart, in one case on
the same day; (ii) the implausible mutation rate that would be needed to explain this
variation as newly arising by mutation; and, (iii) the lack of analysis demonstrating
that the data shows any signal of “measurable evolution”.

---

## [Author Response]

We thank the reviewers’ for their thoughtful comments, which we addressed in full
in the revised manuscript. We first highlight the key aspects of the revision and then
address each point in the response below. We realize that this re-submission has been
delayed: as we explain below this is due to our decision to re-sequence all strains to
substantially greater depth to ensure the reproducibility and validity of our results.
This was critical as this is a resource intended for the wider community.

1) Extensive deeper sequencing of each strain shows high concordance in genotype calls:
the reviewers pointed out that there was a lower than desired agreement between our
Sequenome iPLEX assay genotyping and the corresponding base calls from our sequencing
data (87% in the original study on average per patient, 77% averaged over all samples
with some strains near 100% and others much lower; new Figure 1—figure supplement 1 in red). The reviewer raised the possibility of
a sample switching. We first ruled out the possibility of a sample switch by repeating
our genotyping from newly isolated DNA, but many of the discrepancies remained. We then
examined the discordant calls and found that those were typically related with lower
sequence coverage. Although our initial analysis used best practices at the time of
original data collection, in some regions of lower sequencing coverage our GATK pipeline
defaulted to the genotype of the provided reference genome, resulting in erroneous SNP
calls when comparing between strains. Because our strains are also a resource used
widely by the community, we decided to re-sequence all the strains to high coverage
(53-283X, an average of 103X); this was not feasible when our data was initially
collected, but major increases in sequencing yields made it readily possible now. The
new sequencing data resolved the discrepancies, such that concordance between Sequenom
iPlex and Illumina data reached 94% across all patients (see also the response to the
issue 3 below; Figure 1—figure supplement 1 in blue). All the key results and conclusions that we originally reported
were all reproduced in this deeper analysis.

2) The new data has added many new interesting genes to our list of those most likely to
have adaptive polymorphisms. Our repeated sequencing revealed 4,756 persistent SNPs
across all isolates and 240 recurrent genes. While previous clustering analysis of
persistent and recurrent SNPs failed to produce significant functional enrichments,
clustering of the recurrence matrix of our new data reveals significant functional
enrichment in four clusters (revised Figure 5).
Additionally, of the 240 recurrent genes identified, 23 were available as deletion
library mutants (up from five previously), and we evaluated them for phenotypic
differences (revised Figure 8). These new data
significantly extend our previous manuscript, and have allowed us to make additional
conclusions regarding drug resistance and adaptation to the host environment.

3) We have clarified the two possible evolutionary scenarios—de novo mutation and
selection or selection on largely pre-existing variation. We now clearly note that,
although the data favors selection of pre-existing variation as the predominant mode, we
cannot rule out that some of the observed variation was due to de novo events that
occurred in the sampling window.

4) We have provided statistical modeling where possible, including stochastic
simulations to model the occurrence of (1) persistent non synonymous SNP-containing ORFs
associated with changes in MIC, and (2) the expected degree of recurrence amongst these
ORFs.

5) We have heeded the reviewers’ suggestion to expand our description and
interpretation of the validation experiments with the available deletion mutants for
recurrent loci; as the number of recurrent loci grew, we now tested 23 deletion mutants,
three of which were significantly more fit than the parental strain (SN250) as
determined by growth in standard tissue culture media (RPMI).

6) We assessed the rate and time scale of evolution by applying Bayesian Evolutionary
Analysis by Sampling Trees (BEAST) in the one patient series with a sufficient density
of longitudinal samples and have included that analysis.

7) We addressed all other comments, including clarification of our use of the word
“clonal” as detailed in the point-by-point response below.

Response to specific comments:

*The manuscript by Funt and colleagues provide an in-depth analysis of whole
genome sequence of* C. albicans *progressively in HIV patients to
infer selective events and describe patterns of mutations, and those that survive and
likely have an impact on resistance. Overall, the study is large and the data
collected is very important and relevant with a degree of phenotypic assessment
paired with the sequence data. The experimental and sequence analysis methodologies
seem to be state of the art*.

We thank the reviewers for these comments. As we note below, realizing that the data set
could serve as a gold standard (and indeed is being used by several labs already), we
decided to leverage the advances in sequencing yields, to generate much deeper
sequencing and better analysis. Below we address each of the concerns raised.

*1) One major concern arises from consideration of all sections of the
manuscript. There is very little statistical modeling to assess how likely these
combinations of certain mutational patterns are due to real selective events due to
specific phenotypic consequences or just by chance. Most of the manuscript is
presented in this style, and it is hard to know which signals are truly significant.
We provide two of the many examples in the manuscript to illustrate this
point*:

*In the Results section, the authors state: “We consider… two to
three isolates”. While this is a reasonable argument, without proper
statistics it is not possible to evaluate it. Perhaps some degree of linkage
disequilibrium would be useful to consider*.

In the Results section, the authors also mention: “The recurring
appearance of LOH events that coincide with changes in MIC suggests that they have
been positively selected…” This is a “coincidence”, as
the authors say, but is it significant?

We thank the reviewers for this comment. We appreciate the suggestion for linkage or LD
analysis, however it is important to emphasize that linkage/LD, by their very nature,
are a function of mating and meiosis; and while mating can be engineered in the
laboratory, meiosis has never been observed in *C. albicans.* Instead,
*C. albicans* is thought to proliferate primarily by asexual
reproduction and thought to only rarely undergo a cryptic, parasexual lifestyle in which
two diploid individuals fuse into a tetraploid, and followed by chromosome loss in
subsequent mitotic divisions to a converge on diploidy. This feature of *C.
albicans* biology is a major limitation to genetic dissection of *C.
albicans* population genetics either in the lab or in natural settings, and
is important to bear in mind when considering our work.

It is for this reason that we think that those mutations that are simultaneously
concomitant with increases in drug resistance and that are also recurrent are likely a
footprint of selection (regardless of whether all were selected from existing diversity
within the population, all arose de novo, or perhaps, a combination thereof). Without a
substantially increased number of samples and non-treated human-passaged *C.
albicans* longitudinal isolates, it is infeasible to adequately model this
system for particular point mutations and genes affected. Furthermore, our samples were
archived over decades, by clinicians who have typically archived only a single clone
(sampled from what is likely a complex heterogeneous population). Our assessment of the
larger gene sizes for those genes with recurrence in the original manuscript was an open
acknowledgement of the possible contribution of random mutations to our results.

In the revised manuscript, we have now more clearly described the inherent challenges of
this system and the limitations of our analysis, and we have attempted to clarify its
statistical basis wherever possible. First, we now highlight these challenges early in
the manuscript, clearly acknowledging the inherent limitations at deriving direct
statistical significance driven by a model of the underlying phenomena. Second, whenever
possible, we articulate the statistical basis for our analyses. For example, consider
the LOH presented in Figures 3 and 4. There
are 30 LOH events, if we limit LOH events to chromosomes arms (of which they occur on 15
out of 16 possible chromosome arms). Of these 30 LOHs, 23 meet our standard for
persistence. The right arm of chromosome 3 has a persistent LOH 7 times. If we assume a
naïve binary distribution, then the probability of this occurring is less
than (237)(115)7(1415)16 = 4.76 × 10^-4^. The same calculation
applies to the left of arm 5 (Chr5L), but with 5 LOH events for a probability of 0.0128.
Unlike chromosome 3, Chr5L is known to be a functionally significant LOH ([13]; [72] and [74]). We have added this analysis to the text.

Furthermore, we now use stochastic simulations to generate null distributions of: (1)
the expected number of persistent non-synonymous SNP-containing ORFs associated with
changes in MIC; and (2) the degree of recurrence amongst these ORFs. These models
(described in the revised Methods) suggest that the number of observed cases of each of
(1) and (2) is significantly larger than expected by chance (p<0.0001). The
distributions are shown in Author response image 1 below, and the results are reported
in the revised text.Author response image 1.Null distributions of (a) the expected number of persistent, recurrent non
synonymous SNPs outside of LOH regions, and (b) persistent non-synonymous SNPs
associated with MIC changes outside of LOH regions.

*2) The last part of the Results describes some validation experiments where the
specific mutations are brought to an experimental setting with deletion mutants. This
is an experiment that was necessary but surprisingly it was not given the proper
weight (it was only a small fraction of the total text), plus it was not as
conclusive as one would like regarding the MIC phenotype*.

We thank the reviewers for the suggestion to further emphasize these results . The
diploid genome and lack of a complete sexual cycle make genetic manipulation of
*C. albicans* challenging. Therefore, we relied on an existing
resource of available homozygous deletion mutants from a deletion strain collection of
674 loci to gain insight into the contribution of these loci to drug resistance and
fitness. In the original manuscript, we tested all those loci recurrently mutated by our
original analysis for which a deletion mutant was available.

Following our deeper sequencing, we could reliably detect a larger number of recurrently
mutated loci, and hence could test 23 deletion strains (revised Figure 8). In this expanded panel of tests, consistent with a role
in host adaptation, 8 out of 23 mutants affected in vitro fitness in a culture medium
thought to approximate in vivo conditions. Strikingly, three mutants show an increased
level of fitness relative to the WT parental isolate, SN250. We have clarified the text
to convey this point. The results reported in the original submission were from fitness
experiments conducted using the drug-resistant reference strain (also used in the
fitness experiments illustrated in revised Figure 7). In principle, any reference strain can be used in relative fitness
experiments. However, in this revision we used the *same* reference
strain to quantify the fitness of the clinical isolates and the deletion mutants, such
that that results in Figure 7 and Figure 8 can be readily comparable by the
reader.

*3) Considering the central role in the paper of the various classes of mutations
called from the sequencing reads, I think that one technical issue in particular
deserves further exploration and/or discussion. The SNP calling accuracy was found to
be 87.3% on average (using a Sequenome iPLEX genotyping assay; Results,
“Sequenome iPLEX genotyping assay”, and*
Table 2*). There was
however an important inter-sample variability, with very poor results observed in
half of the tested series (<65% calling accuracy). This is worrying. Could this
be due to intra-patient diversity, with different* Candida
*subpopulations being sampled for the two assays? Is there any other potential
explanation? If not, these inconsistencies should be listed as an important
limitation of the study.*

The same genomic DNA from a single archived clone was used for both the original
sequencing and the Sequenom assay. We were originally concerned that a switching in
samples was the cause of the discrepancy, but repeated genotyping and re-examination of
the Sequenome iPLEX versus the sequencing data suggested that when sequence coverage was
relatively low, our GATK pipeline defaulted to the genotype of the provided reference
genome resulting in erroneous SNPs calls across isolates in the same time course. Given
that the extensive sequence data set is a major strength of our study, and with the
substantial increase in sequencing yields since the time when we originally sequenced
these isolates, we chose to re-sequence all the strains to high coverage (53-283X,
average depth of coverage: 103X).

The new sequencing data resolved the discordance between the Sequenom and sequencing
genotype calls (Table 2) such that our average
concordance rose to 94% (85%-98%, Figure 1—figure supplement 1). To further understand the nature of our
discordant sites, and importantly, determine if improved filtering of the SNP calls
could improve concordance, we examined the discordant sites for any potential base bias
(Figure 1—figure supplement 1). From
this analysis, it is apparent that the majority of discordant sites are those that are
identified as homozygous by Sequenom iPlex (Figure 1—figure supplement 1, teal bars), but as heterozygous by Illumina
sequencing (Figure 1—figure supplement 1, orange bars). We compared several quality metrics for these sites in our
Illumina data for discordant vs. concordant sites (Figure 1—figure supplement 1: depth of coverage (DP), mapping
quality (MQ), PHRED scaled quality scores (QUAL), the QD score (confidence of a variant
call), and the ratio of alternative allele to reference allele (AB)). We found that
while the distributions of some quality scores for discordant sites are somewhat lower,
the degree of overlap is so extensive that any improvement in specificity will come at a
substantial sensitivity cost. Given that the discordance rate is already low, and that
we focus on SNPs that both persist within a series and recur across series (and are thus
called many independent times), we opted not to use any stricter thresholds for SNP
calling. We present this comparison in the revised text, in the Results section.

Importantly, all previous conclusions were reproduced in this enhanced analysis, and we
were able to draw additional conclusions based on these higher quality data. For
example, the new set of 240 mutated genes is now enriched for several functions known to
influence pathogenicity, including fungal-type cell wall (18 genes, p<0.0012) and
cell surface genes (24 genes, p<0.00012), with several members in each of three
cell wall gene families important for biofilm formation and virulence (34): the Hyr/Iff proteins (HYR1
and 3, IFF8 and 6), the ALS adhesins (ALS1-4,7,9), and the PGA-30-like proteins (7
genes) ([Supplementary-material SD4-data]). All three families are specifically expanded in the genomes of
pathogenic *Candida* species (8). In addition, seven members of the FGR gene family, involved in
filamentous growth that is specifically expanded in *C. albicans* (8), are also among the 240 genes
([Supplementary-material SD4-data]).

*4) The paper is presented as a study of the within-host mutations that occur and
are selected for in the* Candida albicans *genome between
longitudinally sampled genomes before and after the introduction of antibiotic
treatment. However, the data suggest that the variants that distinguish* C.
albicans *genomes at different time points did not arise during the sampling
window but were largely, maybe entirely, pre-existing (standing variation). This view
is based on: (i) the very considerable genetic diversity (thousands of SNP
differences) between isolates of the same strain within the same host, sampled less
than a few months apart, in one case on the same day; (ii) the implausible mutation
rate that would be needed to explain this variation as newly arising by mutation;
and, (iii) the lack of analysis demonstrating that the data shows any signal of
“measurable evolution”.*

We completely agree with the reviewers that, given the evidence of considerable genetic
diversity existing in the same host, selection may primarily be driving an increase in
the frequency of variants that already existed in the population. We have revised the
text throughout the manuscript to clarify this point (especially in the Introduction and
Results sections). Nevertheless, there is also ample evidence that mutation rates are
increased under stressful conditions (e.g., drug treatment) in many systems, including
yeasts, akin to what occurs in cancer (26). Mitotic recombination events that give rise to LOH occur at a much
higher rate (3 x 10^-5^) than single-nucleotide mutation rates and are also
increased by stress. Specifically relevant here, LOH rates are elevated in *C.
albicans* treated with fluconazole (20). Thus, we cannot rule out the possibility that some genetic
changes, including LOH events, occurred de novo during the treatment and sampling
window. We discuss this point in the revised manuscript, in the Discussion.

Importantly, we are unable to distinguish pre-existing from de novo events with the data
in hand. The rate of collection and number of samples we have for the majority of
patients likely does not constitute a dataset that can be analyzed by existing
approaches to ‘measurably evolving population’ (MEP, as per Drummond et
al., 2003), and we do not want to overstate our evolutionary scenario. However, even if,
as the reviewers suggest, *“all ‘persistent mutations’ are
those pre-existing variants driven to sufficiently high frequency by selection to
ensure repeated sampling, whereas transiently observed variants are
not”,* this scenario is still an evolutionary process, involving
heritable variation and selection, and can still inform our functional understanding of
host-pathogen interactions.

With these limitations in mind, we were intrigued by the reviewers’ suggestion to
assess the rate and time scale of evolution in these strains. While the majority of our
patient series represent only a few sequentially isolated clonal samples, Patient
1’s series consists of 16 clonal, sequential isolates sampled over a period of
nearly two years, and thus these samples can be analyzed for evidence of measurable
evolution. We have used a Bayesian Markov Chain Monte-Carlo approach (Bayesian
Evolutionary Analysis by Sampling Trees, BEAST; Drummond et al., 2012) to assess the
rate of mutation in these populations and the likely time to most recent common ancestor
(TMRCA).

To test the hypothesis that the majority of mutations occurred before the sampling
period, we sampled trees generated using a set of parameters based on available data.
Because there are no estimates of the per-basepair, per-generation mutation rate in
*C. albicans*, we began with the assumption that *C.
albicans* mutates at a rate similar to the per-base pair, per-generation rate
of *S. cerevisiae* (Lynch et al., 2008). We then adjusted this rate to a
per-day rate based on the generation time observed for C*. albicans* in a
mouse model (1.56 generations per day, [23]), and determined the TMRCA for the isolates from Patient 1. Based on
these assumptions, the TMRCA for these 16 isolates is ∼ 235 years (Author
response image 2A). Indeed, using these parameters, many of the nodes separating two
isolates are dated to be ∼150 years in the past. While little is known of the
genetic diversity present *in C. albicans* populations at the time of
colonization, it seems unlikely that the most recent common ancestor for the clonal
population seen in Patient 1 evolved 235 years prior to isolation: if colonization
occurs primarily during birth, it would mean that much of the diversity present in this
patient actually evolved in the patient’s grandmother, or great grandmother.

To better understand the diversity present in the population, we repeated this analysis,
but with a broader estimate of mutation rate (for the prior, the same mean as above, 5.5
x 10^-10^ mutations/bp/day, but with a larger standard deviation), resulting in
a 5 to 55 fold increase in the estimated substitution rate, and a corresponding
reduction in the likely TMRCA (Author response image 2B,C). To determine which of these
three models best fit the data, we performed a pairwise calculation of Bayes factors
(Author response image 2D), and we find that model (C), with an estimated substitution
rate of 8.48 x 10^-8^ best approximates our data. These results are consistent
with the reviewer’s comment that considerable variation developed before the
onset of treatment and sampling; however, they also suggest that a substantial portion
of the observed variation occurred during the treatment period in Patient 1.
Importantly, they also suggest that an elevated mutation rate likely explains some of
the diversity we observe, consistent with previous observations of in vivo
hypermutability during infection and treatment (Oliver et al, Science 2000, [26], [20]).Author response image 2.The results of three separate MCMC analyses based on differing priors for
mutation rate are shown. This analysis is based only on SNPs from isolates of
Patient 1.

Further work is necessary to determine: (1) the extent of population diversity at any
given time point, (2) the extent of the bottlenecks during colonization and drug
treatment, and (3) the impact of the host immune response and antifungal therapy on the
mutation rate *Candida albicans*. In light of these issues—which
are relevant to a variety of pathogens—this analysis is particularly provocative.
However, as many of the parameters necessary to make these estimates are poorly informed
and our sampling limits the analysis to only one patient series, we do not feel this
analysis merits inclusion in the manuscript.